



# Technical Note: assessing predicted cirrus ice properties between two deterministic ice formation parameterizations

Colin Tully , David Neubauer , and Ulrike Lohmann

Institute for Climate and Atmospheric Science, ETH Zurich, Zurich, Switzerland

**Correspondence:** Colin Tully (colin.tully@env.ethz.ch) or Ulrike Lohmann (ulrike.lohmann@env.ethz.ch)

**Abstract.** Determining the dominant ice formation mechanism in cirrus is still an open research question that impacts the ability to assess the climate impact of these clouds in numerical models. Homogeneous nucleation is generally well understood. There is more uncertainty surrounding heterogeneous nucleation processes due to the complex physio-chemical properties of ice nucleating particles (INPs). In addition, determining whether a heterogeneous nucleation process follows a time-dependent
(stochastic) or a time-independent (deterministic) approach increases the level of complexity in numerical modeling applications. Kärcher and Marcolli (2021) introduced a new deterministic ice formation parameterization based on the differential activated fraction (AF), arguing that with explicit INP-budgeting this approach could help correct a potential over-prediction of the importance of heterogeneous nucleation within cirrus. We formulated a general circulation model (GCM)-compatible version of the differential AF parameterization and compared it to the method currently employed in the ECHAM6.3-HAM2.3
GCM that is based on cumulative AF, with implicit INP-budgeting. In a series of box model simulations that were based on the cirrus sub-model from ECHAM, we found that the cumulative approach likely under-predicts heterogeneous nucleation in cirrus as it does not account for INP concentration fluctuations across GCM timesteps. However, as the cases that we simulated in the box model were rather extreme, we extended our analysis to compare the differential and cumulative AF approaches in two simulations in ECHAM-HAM. We find that choosing between these two approaches impacts ice nucleation competition
within cirrus in our model, but the climate impact is small and insignificant based on our five-year simulations. We argue that while our GCM-compatible differential AF parameterization is closer to first principles, the default approach based on cumulative AF is simpler and leads to more interpretability of the climate model results.

## 1 Introduction

Historically, clouds introduced the largest uncertainties in the projections of climate. Today, however, significant improvements
in our understanding provide more confidence that cloud feedbacks will amplify climate warming in the future, for example through reductions in tropical low cloud cover (Zelinka et al., 2017, 2020; Forster et al., 2021). While more is understood about cloud feedbacks in response to a changing climate state (e.g. from the forcing associated with a quadrupling of atmospheric $CO_2$ as simulated by CMIP6 experiments Eyring et al., 2016), there is less understanding of the present-day radiative forcing from the interactions between clouds and aerosol particles that can act as cloud condensation nuclei (CCN) and INPs (Heyn
et al., 2017; Storelvmo, 2017; Bellouin et al., 2020). In fact, estimates of the present-day effective radiative forcing due to



aerosol-cloud interactions, which includes rapid adjustments (Sherwood et al., 2015), (e.g. a cloud glaciation effect due to elevated INP concentrations, Lohmann, 2002), are in the range of -1.36 [-2.65 to -0.07] $\mathrm{Wm}^{-2}$ for an average period of 2005 to 2015 relative to 1850 (Bellouin et al., 2020), or –1.0 [–1.7 to –0.3] $\mathrm{Wm}^{-2}$ for 2014 relative to 1750, based on CMIP6 experiments, as reported in the latest assessment report by the Intergovernmental Panel on Cliamte Change (IPCC), (Forster
et al., 2021).

The interactions of aerosols with liquid clouds are mostly well established (Twomey, 1959, 1977; Albrecht, 1989; Ackerman et al., 2004; Small et al., 2009; Christensen et al., 2020), whereas the impacts of aerosols acting as INPs on mixed-phase and ice clouds (i.e. cirrus) contain a higher level of uncertainty (Storelvmo, 2017; Bellouin et al., 2020). For cirrus, the subject of this technical note, accurately simulating ice formation mechanisms is still an open research topic that impedes further
understanding of cirrus climate impacts, including assessing potential climate intervention strategies (Storelvmo et al., 2013; Zhou and Penner, 2014; Penner et al., 2015; Gasparini and Lohmann, 2016; Krämer et al., 2016; Kärcher, 2017; Storelvmo, 2017; Gasparini et al., 2020; Krämer et al., 2020; Kärcher et al., 2022; Tully et al., 2022c).

Cirrus form from the nucleation of ice in the upper troposphere via homogeneous or heterogeneous nucleation. The prevalence of one mechanism over the other in the atmosphere remains uncertain, leaving a large gap in understanding cirrus
radiative properties as these two ice formation mechanisms can lead to vastly different cloud properties (Lohmann et al., 2008; Storelvmo, 2017; Krämer et al., 2020; Cziczo et al., 2013). Homogeneous nucleation occurs as the spontaneous freezing of aqueous solution droplets (otherwise referred to as liquid aerosols) at conditions with a temperature below roughly 238 K and a high supersaturation with respect to ice (Koop et al., 2000). Ice crystal growth following such an event is typically limited as water vapor that is sparsely populated in the upper troposphere is rapidly consumed (Ickes et al., 2015), leading to numer-
ous and small ice crystals that act to absorb outgoing longwave (LW) radiation and re-emit it at a lower magnitude than the underlying surface. Heterogeneous nucleation occurs at a much lower ice supersaturation and at warmer temperatures than homogeneous nucleation due to the presence of an INP surface. Several modeling studies found that a sufficient number of INPs can inhibit homogeneous nucleation through preferential formation of ice crystals followed by rapid deposition of water vapor onto their surfaces (Lohmann et al., 2008; Storelvmo et al., 2013; Storelvmo and Herger, 2014; Storelvmo et al., 2014; Penner
et al., 2015; Jensen et al., 2016). As the number of ice crystals in this case is rather limited by the availability of INPs, which are also sparsely populated in the upper troposphere (DeMott et al., 2003), they tend to be larger in size than those formed purely by homogeneous nucleation. This leads to an optically thinner cloud that is less effective at trapping LW radiation in an effect coined the "Negative Twomey effect" (Kärcher and Lohmann, 2003). However, under high dynamic forcing (e.g. high vertical velocity) or without a sufficient number of INPs, heterogeneous nucleation may not be efficient enough to prevent high
ice superaturations required for homogeneous nucleation.

The theory behind homogeneous nucleation is relatively well understood (Koop et al., 2000; Ickes et al., 2015), with new evidence perhaps suggesting higher freezing onsets at cold temperatures for sulphuric acid droplets (Schneider et al., 2021). However, heterogeneous nucleation in general is still a topic of substantial research (Cziczo and Froyd, 2014; Kanji et al., 2017). Specifically, the ability of certain materials to act as an INP, e.g. on mineral dust (Murray et al., 2012) or on black
carbon particles (Mahrt et al., 2018, 2020), as well as the characterization of their abundance in the atmosphere (Li et al.,



2022). Furthermore, heterogeneous nucleation can occur via several processes. For example, from immersion freezing within a solution droplet or by the deposition of water vapor onto the INP surface (Vali et al., 2015; Kanji et al., 2017; Heymsfield et al., 2017), the former of which is thought to be the most common heterogeneous nucleation mechanism in cirrus (Kärcher and Lohmann, 2003).

In general, as there are more factors that govern the complexities of heterogeneous nucleation than homogeneous nucleation, it is rather difficult to simulate this mechanism in models, as well as assess its impacts on ice nucleation competition within cirrus. Due to their coarse resolution, GCMs rely on parameterizations of heterogeneous nucleation mechanisms that are based on laboratory measurements of ice formation. A common method to simulate the number of ice crystals that form from a heterogeneous nucleation event is based on the AF of available INPs. This quantity is derived from cloud or continuous flow

chamber experiments of the number of frozen particles (Kärcher and Marcolli, 2021). Vali (1971) and Vali (2019) defined two approaches for determining the number of INPs that become ice active. The differential AF approach describes the number of INPs that are active within a certain temperature interval (assuming temperature decreases during a freezing experiment or as a theoretical air parcel rises within a model), whereas the cumulative AF describes the total number of active INPs between the temperature at the onset of ice activity and a given (lower) temperature, which equates to the integral of the differential

AF over the specified temperature range. As Vali (1971) explains, on the one hand, the former approach is useful to describe the freezing behavior of a single particle. On the other hand, the latter approach can be used to determine the ice nucleation ability of a population of particles, which is relevant to modeling cirrus ice formation processes that can occur on numerous particles. However, as Kärcher and Marcolli (2021) highlight, care must be taken when determining which approach to use when calculating the number of ice crystals that can form on INPs. For example, if a model explicitly removes INPs from the

total available population after each ice formation event and adds them to the newly nucleated ice number concentrations (i.e. "INP-budgeting"), then using the cumulative AF approach, which is based on the total (integrated) number of active INPs, may overpredict the number of heterogeneously nucleated ice crystals.

Kärcher and Marcolli (2021) introduced a new parameterization to simulate the number of ice particles resulting from a heterogeneous nucleation event that is based on the differential AF (Vali, 1971, 2019) when employing an INP-budgeting

approach (Section 2.1). This method demonstrated that it is able to counteract the potential over-prediction of heterogeneous nucleation in cirrus. Meanwhile, Muench and Lohmann (2020) reformulated the ice nucleation mechanisms for cirrus in the ECHAM-HAM GCM (Stevens et al., 2013; Neubauer et al., 2019; Tegen et al., 2019) to also include an AF-based approach that is, instead, based on cumulative AF. Note, this new approach in ECHAM-HAM is not the same as the cumulative AF approach described by Kärcher and Marcolli (2021), as Muench and Lohmann (2020) introduced implicit INP-budgeting by

using a differential ice crystal number concentration (ICNC) variable that accounts for the issue stated by Kärcher and Marcolli (2021).

As the differential AF method introduced by Kärcher and Marcolli (2021) was applied in a process model, it does not capture the complexities of the cirrus formation environment like in a GCM. Therefore, a technical analysis is needed for the implications of using a new approach to simulate deterministic ice nucleation processes via AF. In this technical note, we

present a comparative analysis of cirrus ICNC between a GCM-compatible differential AF parameterization based on Kärcher





and Marcolli (2021) and the default AF approach used in ECHAM-HAM (Muench and Lohmann, 2020). In Section 2 we describe the box model we developed based on ECHAM-HAM to analyze these differences followed by our results and a discussion in Section 3. We finish with some concluding remarks in Section 4.

## 2 Methods

We formulated a box model to analyze deterministic, AF-based ice nucleation within cirrus clouds. The model is based on the cirrus ice nucleation scheme in the ECHAM-HAM GCM by Kärcher et al. (2006), Kuebbeler et al. (2014), and Muench and Lohmann (2020). In this note, we utilize the box model to compare a GCM-compatible differential AF approach based on Kärcher and Marcolli (2021) for heterogeneous nucleation to the AF approach by Muench and Lohmann (2020), hereafter abbreviated as ML20.

### 2.1 Ice formation mechanisms

The cirrus model in ECHAM-HAM is called from the cloud microphysics scheme and calculates the number of new ice crystals that form in in-situ cirrus. It uses a sub-stepping approach to simulate the temporal evolution of the ice saturation ratio in an adiabatically ascending air parcel during the formation stage of a cloud (Kuebbeler et al., 2014; Tully et al., 2022c). In this note, we extracted only the cirrus sub-model code from ECHAM-HAM to formulate a box model of ice nucleation within cirrus, see Section 2.2 and Section 2.3. In its full form, the cirrus sub-model calculates the competition of water vapor between deposition onto pre-existing ice particles and phase transitions by homogeneous or heterogeneous nucleation processes that form new ice crystals (Tully et al., 2022c). Muench and Lohmann (2020) introduced a new method to more easily distinguish between new ice formation mechanisms by categorizing them by either threshold or continuous processes.

Threshold processes are based on the stochastic nature of nucleation rates, and include homogeneous nucleation of liquid sulphate aerosols and immersion freezing by internally mixed (soluble) mineral dust particles. As its name suggests, as soon as the ice saturation ratio ($S_i$) reaches a critical value, the model assumes nucleation rates are efficient enough such that all of the available aerosols nucleate ice during a single timestep of the cirrus sub-model. For immersion freezing of internally mixed mineral dust particles, it is assumed that only 5 % of the background concentration can act as INPs (Gasparini and Lohmann, 2016; Muench and Lohmann, 2020). The continuous processes are deterministic (time-independent, Kärcher and Marcolli (2021)), and are based on laboratory measurements of AF, which are determined by temperature and $S_i$. In the cirrus sub-model, continuous processes include deposition nucleation onto externally mixed (insoluble) mineral dust particles only, following laboratory-based measurements of AF by Möhler et al. (2006).

### 2.1.1 KM21 differential AF approach

Kärcher and Marcolli (2021), hereafter KM21, introduced a new method to describe freezing processes by the number of activated particles. In their study, they argue that using laboratory-based cumulative AF ($\phi$) when coupled to INP budgeting, over-predicts the number of ice crystals originating from heterogeneous nucleation as $\phi$ is based on the total INP population





(N$_0$). Instead, they formulated a "differential AF" ($\psi$) approach, which considers only the number of particles that can activate as a result of incremental changes of S$_i$ during a timestep $j-1$ to $j$. The method is based on the probability that the remaining INPs in the current timestep $j$ do not become ice-active. Thus, $\psi$ follows the form (Equation 6 of KM21):

$$\psi_j = \frac{\Delta\phi_j}{1-\phi_{j-1}}, \text{where} \quad \Delta\phi_j = \phi_j - \phi_{j-1}, \text{and} \quad 0 \leq \psi_j \leq 1 \tag{1}$$

As a short conceptual example of their argument (see also KM21 Figure 1), starting from an initial INP population (N$_0$) of 100 L$^{-1}$, in the first cirrus model timestep $\phi_{j=1}$ is calculated as 0.05 under ambient temperature and S$_i$. Therefore, the resulting ICNC after the first timestep is 5 L$^{-1}$, which equates to $\Delta N = 5$ L$^{-1}$ INPs. With INP budgeting, the resulting population after the first timestep is N$_0$-$\Delta N = 95$ L$^{-1}$. In the second timestep $\phi_{j=2}$ is calculated as 0.1. Using this value alone results in a $\Delta N$
$= 9.5$ L$^{-1}$, and thus a total ICNC after this step of 14.5 L$^{-1}$. However, as $\phi$ is based on N$_0$, the total ICNC should be 10 L$^{-1}$ in the second timestep, therefore the number of activated particles is over-predicted in this case as the INPs activated in the first timestep were ignored during activation in the second timestep. Using the differential AF approach as presented in Equation 1, with $\phi_j = 0.1$ and $\phi_{j-1} = 0.05$, the resulting $\psi_j$ equates to roughly 0.05. When applying this to the number of available INPs (95 L$^{-1}$), $\Delta N = 5$ L$^{-1}$, bringing the total ICNC after the second timestep instead to 10 L$^{-1}$. Although the resulting ICNC
values after the second timestep in this short example are not large, not accounting for previously activated INPs in a correct manner could drastically increase the amount of heterogeneous nucleation on mineral dust particles, leading to vastly different cirrus properties.

The KM21 method in its current form only considers one cirrus formation cycle, which in ECHAM-HAM occurs as a sub-loop within a single GCM timestep of 7.5 minutes. A typical cirrus clouds exists over several GCM timesteps. Between each
timestep, not only can the number of available INPs in a given gridbox differ based on aerosol transport and vertical diffusion, but also the temperature and S$_i$ conditions can change based on the dynamics of the model. Therefore, we made adjustments to the KM21 approach presented above for climate model compatibility (KM21_GCM). The new approach is described in more detail in Section 2.2.

### 2.1.2 Default ML20 cumulative AF approach

Explicit INP budgeting is not considered for dust deposition nucleation in the default version of the cirrus sub-model in ECHAM-HAM. Instead, budgeting is implied at each cirrus sub-timestep ($j$) through the differential ICNC ($\Delta$ICNC), following ML20, that takes the form:

$$\Delta ICNC_j = \phi_j \cdot N_0 - ICNC_{j-1} \tag{2}$$

where $\phi_j$ is the AF based on Möhler et al. (2006), N$_0$ is the initial INP population, and ICNC$_{j-1}$ is the ICNC from the
previous cirrus sub-model timestep. Negative $\Delta$ICNC values are set equal to zero. Therefore, the ICNC at each cirrus sub-model timestep is only ever updated if the new ice formation exceeds the previously formed concentration. The advantage of this approach over KM21 is that it is simple (see Section 2.2). The activation of INPs during the lifetime of a cirrus is implicitly included by requiring that ICNC$_j$ > ICNC$_{j-1}$.





## 2.2 Cirrus box model

As described previously, we formulated a box model based on the cirrus sub-model in ECHAM-HAM. It simulates the temporal evolution of $S_i$ during the adiabatic ascent of a theoretical air parcel. $S_i$ evolves through the balance between the updraft velocity, which is used as an input parameter to the sub-model, and a fictitious downdraft that quantifies the effect of vapor deposition onto newly-formed or pre-existing ice crystals (Tully et al., 2022c). This quantity is termed the "effective updraft velocity". If an environment has a high background INP concentration that leads to numerous new ice crystals forming via

heterogeneous nucleation, and/or if it contains a high concentration of pre-existing ice crystals (e.g. from convective detrainment), then the vapor deposition onto these ice crystals may be sufficient to prevent the development of high $S_i$ values suitable for homogeneous nucleation.

To emulate the GCM we define starting conditions for temperature, pressure, $S_i$, and the updraft in order to simulate the adiabatic ascent of an air parcel during the cirrus formation stage. To simulate deterministic, AF-based ice formation onto

externally mixed accumulation and coarse mode mineral dust particles (Stier et al., 2005; Zhang et al., 2012), we also defined two "freezing modes", respectively, following Muench and Lohmann (2020). Full ice nucleation competition (Gasparini and Lohmann, 2016; Tully et al., 2022c) was also tested by adding additional modes for homogeneous nucleation of liquid sulphate aerosol, immersion freezing of internally mixed mineral dust particles, and pre-existing ice. However, the results for these latter tests are not shown in this note as the competition between ice formation mechanisms as well as vapor deposition onto pre-

existing ice crystals did not change the outcome of our box model compared to our dust deposition-only tests. The starting conditions we tested for this study are described in Section 2.3.

The KM21 approach was introduced in Section 2.1. As this method is only valid for a single cirrus formation cycle, we reformulated the parameterization in our box model for compatibility with multiple cirrus cycles in a GCM (KM21_GCM). Specifically, in order to calculate the differential AF ($\psi$) in the cirrus sub-model after the first GCM timestep ($i > 1$), following

Equation 1, the final AF ($\phi_{j=n}$) from the cirrus cycle in the previous GCM timestep ($i$-1) is required, where $n$ is the number of cirrus sub-model timesteps. As a result, we implemented a $\phi$ tracer in our box model that accounts for $S_i$ oscillations to mimic tracing across GCM timesteps. Following KM21, $\phi$ is set equal to the maximum AF reached within a cirrus formation cycle. If in the next cycle the $S_i$ is lower, then $\phi$ remains at the higher value and no new ice formation can occur until the $S_i$, and by extension the AF, exceeds the maximum value reached in the previous cycle.

$S_i$ oscillations are not the only factor to consider across GCM timesteps. INP concentrations can also change. Therefore, an additional tracer for the previous INP concentration ($N_{0,i-1}$) was also implemented in our box model. Note, that both the maximum previous cirrus cycle AF tracer and the previous INP concentration tracer are zero outside of the cloud. Considering these new tracers for the maximum AF ($\phi$) and the previous INP concentration, we reformulated the KM21 calculation of differential AF ($\Psi$, KM21_GCM) as the weighted average of $\phi$ and $\psi$ from the INP concentration ($N_{0,i}$) of the current GCM

timestep (current cirrus formation cycle) as follows:

$$\Psi_j = \frac{\phi_j \cdot (N_{0,i} - N_{0,i-1}) - (N_{0,i-1} \cdot \psi_j)}{N_{0,i}} \tag{3}$$

where $\psi_j$ is the differential AF according to Equation 1. The previous AF ($\phi_{j-1}$) in this case is $\phi_{i-1,j=n}$.



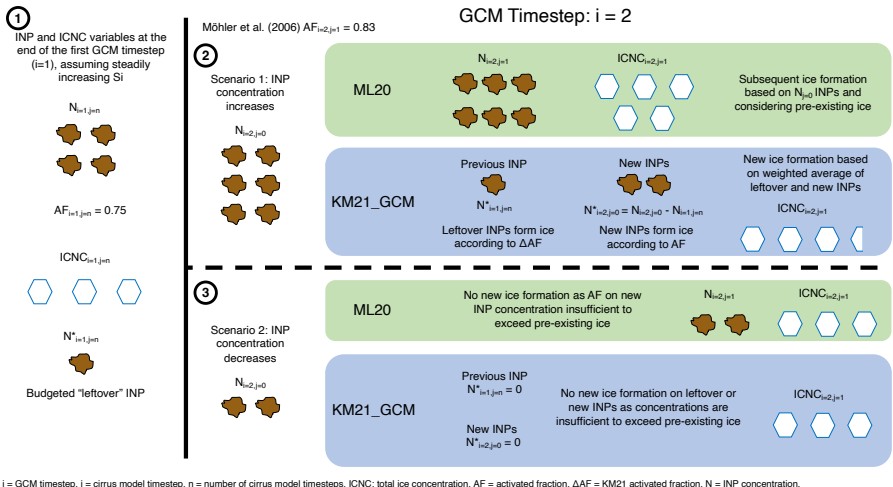

**Figure 1.** Schematic of two illustrative examples of how new INP concentrations are considered in the second GCM timestep ($i = 2$) for the ML20 approach and our newly formulated KM21_GCM approach.

Fig. 1 presents two illustrative examples of how this new method works in our box model compared to ML20. At the end of the first GCM timestep the maximum AF is 0.75 ($\phi_{i-1,j=n}$), and the $\text{ICNC}_{i-1,j=n} = 3\,\text{L}^{-1}$, which formed out of a total INP concentration of $N_{0,i-1} = 4\,\text{L}^{-1}$. In the next GCM timestep ($i = 2$), the INP concentration increases to $N_{0,i} = 6\,\text{L}^{-1}$ for Scenario 1. In the first cirrus sub-model timestep, we assume the $S_i$ increases such that the AF based on Möhler et al. (2006) is 0.83. For the ML20 approach, which considers the current INP concentration ($N_{0,i}$) and the pre-existing ICNC, new ice formation follows the $\Delta$ICNC approach in Equation 2 and the total ICNC increases to $5\,\text{L}^{-1}$. Assuming $S_i$ continues to increase in subsequent cirrus sub-model timesteps, the ICNC following the ML20 approach will increase only if it exceeds the pre-existing ICNC.

KM21_GCM follows a different approach. In order to account for leftover INPs as well as INPs that are still included in ice crystals from the previous GCM timesteps, and before we start the time loop for the cirrus box model, we scale $N_{0,i}$ and $N_{0,i-1}$ by removing the ICNC that formed previously ($\text{ICNC}_{i-1,j=n}$). This allows us to properly consider changes in INP concentrations and $S_i$ across GCM timesteps. In the case of Scenario 1 in Fig. 1, the number of available INPs equates to $3\,\text{L}^{-1}$ after scaling. Out of these INPs, $1\,\text{L}^{-1}$ was leftover from the previous GCM timestep and should nucleate ice following $\psi$, whereas the effective newly available INP concentration, equating to the difference between the scaled INP concentration and the leftover INP concentration, is $2\,\text{L}^{-1}$ and should follow $\phi$ in the first step of the cirrus sub-model. Therefore, the number of ice-active INPs is calculated as the weighted average of these two INP concentrations out of the total scaled INP concentration ($3\,\text{L}^{-1}$) following Equation 3.

No new ice formation occurs under Scenario 2 in Fig. 1 following both the ML20 and the KM21_GCM approaches. On the one hand, even with the larger AF in the second GCM timestep, the number of ice crystals that can form following ML20 on the smaller number of available INPs is insufficient to exceed the pre-existing ICNC. On the other hand, for KM21_GCM,





**Table 1.** Summary of the different trends for large-scale $S_i$ and aerosol concentration that are used as input to the cirrus box to compare ICNC between ML20 and KM21_GCM.

| Trend | Large-scale $S_i$ | INP concentration ($L^{-1}$) |
|---|---|---|
| Increasing | 1.2, 1.3, 1.4 | 2000, 4000, 6000 |
| Decreasing | 1.4, 1.3, 1.2 | 6000, 4000, 2000 |
| Intermediate drop | 1.2, 1.0, 1.4 | 2000, 1000, 6000 |
| Constant | 1.2, 1.2, 1.2 | 2000, 2000, 2000 |

scaling the available INPs by $ICNC_{i-1,j=n}$ equates to zero. As there are no INPs available, the AF is simplified to the KM21 differential AF following Equation 1. As this is applied to zero INPs, there is no new ice formation in this case either.

Note that the two examples in Fig. 1 are simplified cases for illustrative purposes and are non-exhaustive of the changing conditions across GCM timesteps.

### 2.3   Experimental setup

There are several parameters that are available as input to our box model, including updraft, temperature, pressure, large-scale $S_i$, and INP concentration. For simplicity, we tested different combinations of large-scale $S_i$ and INP concentrations over three
cirrus formation cycles that emulate three GCM timesteps. Each combination defines a "trend" that could be expected in a GCM over the three timesteps that we used to conduct our simulations. We tested four different trends for a total of 16 tests with our cirrus box model to compare ML20 and KM21_GCM. The different combinations are summarized in Tab. 1. In these tests, we only considered heterogeneous nucleation on mineral dust, following the AF approaches as described in Section 2.1 and Section 2.2. We assess each case by the error between the KM21_GCM approach and the ML20 approach, relative to
the latter. Finally, we conducted two simulations with the ECHAM-HAM GCM to compare ICNC fields and cloud properties between ML20 and KM21_GCM.

### 3   Results and discussion

### 3.1   Cirrus box model simulations

Of the 16 tests we conducted, six show agreement between ML20 and KM21_GCM in the predicted ICNC, as denoted by
the red shading in Fig. 2. We find agreement for the cases that the large-scale $S_i$ and INP concentration trends follow the same pattern, with the exception of the intermediate drop scenario. We also find agreement for all cases with a constant INP concentration across the three timesteps, except for the case with an intermediate drop in large-scale $S_i$. The disagreements (i.e. non-zero errors) we find with these two exceptions are small ($< 1.0\%$). Nevertheless, we discuss these cases in more detail below. Finally, there is also agreement for the case with a constant large-scale $S_i$ and decreasing INP concentration trend. Note,
in all cases with non-zero error, KM21_GCM predicts higher ICNC than ML20.





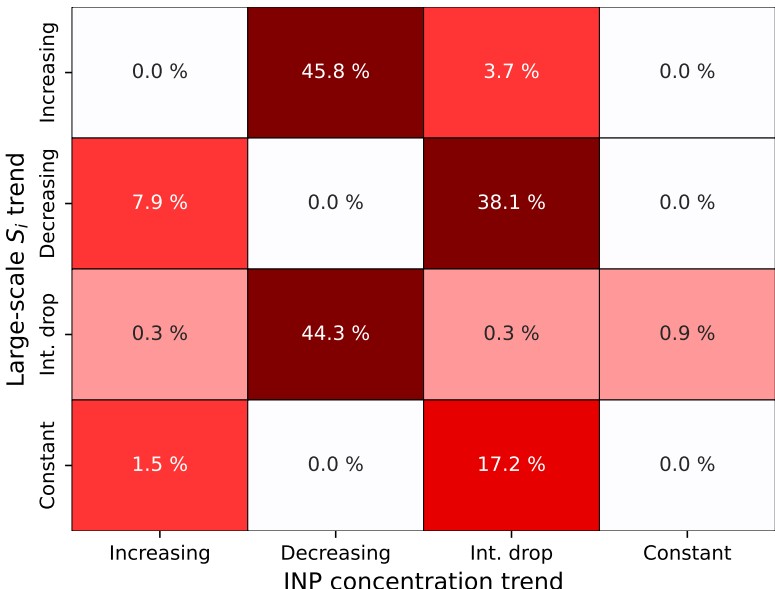

**Figure 2.** Heat map of maximum relative error between KM21_GCM and ML20 in the predicted ICNC from heterogeneous-only ice nucleation for the 16 tests we conducted with different combinations of large-scale $S_i$ and INP concentration trends with the box model. Darker red shading denotes a larger non-zero error. "Int. drop" stands for "intermediate drop".

Agreement between the ML20 and KM21_GCM approaches in the predicted ICNC is expected. On the one hand, while ML20 considers only the starting INP concentration ($N_0$) and does not include explicit INP-budgeting, the ICNC is updated only if the amount of new ice formation as a portion of $N_0$ exceeds the ICNC from the previous sub-timestep in the cirrus box model. As $S_i$ increases within the updraft and the number of INPs that can nucleate ice increases based on higher $S_i$-dependent

AF, eventually all available INPs will become ice-active and $\Delta$ICNC will equate to zero, ceasing all new ice formation, as the AF cannot exceed 1.0. On the other hand, KM21_GCM follows an explicit INP-budgeting approach. Therefore, the number of newly-nucleated ice crystals is proportional to a smaller number of available INPs that, in turn, is based on the number of newly formed ice crystals in each sub-timestep in the cirrus box model. In the case of decreasing $S_i$ and aerosol concentration trends, the lower $S_i$ leads to lower AF values that prevent new ice formation after the first cycle (GCM timestep). While these scenarios

are valuable to understand consistency between ML20 and KM21_GCM, it is useful to examine non-zero errors between the two approaches. Therefore, the rest of this section will focus on the cases where we found disagreements between ML20 and KM21_GCM in the predicted ICNC. For brevity, only a few selected cases where we found non-zero errors are presented in this note. The remainder of the tests we conducted are presented in Appendix A for completeness.

  The largest errors of 45.8 % and 44.3 % occur for the cases with a decreasing INP concentration trend, with an increasing and

intermediate drop in large-scale $S_i$, respectively (Fig. 2). The predicted ICNC and $S_i$ profiles for these two cases are presented





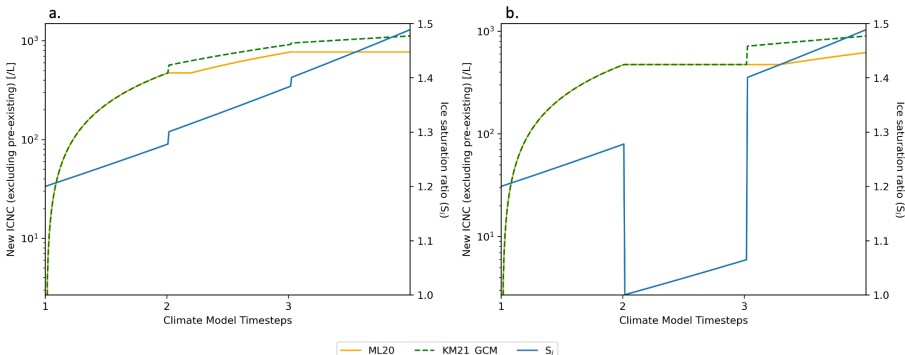

**Figure 3.** Temporal evolution of predicted heterogeneous-only ICNC and $S_i$ for the cases with decreasing INP concentrations across three GCM timesteps for (a) the increasing large-scale $S_i$ and (b) the intermediate drop in large-scale $S_i$. Each line as noted in the legend refers to the predicted ICNC following our ML20 approach (orange solid), our GCM compatible differential AF approach, KM21_GCM, based on KM21 (green dashed), and the $S_i$ (blue solid) based on the large-scale $S_i$ provided as input to our cirrus box model, which evolves according to the effective updraft velocity. Note the difference in scales between (a) and (b) for the predicted new ICNC on the left y-axis of both plots.

in Fig. 3. Non-zero error between the predicted ICNC for KM21_GCM and ML20 occurs from the start of the second cirrus cycle in the first case (Fig. 3a), and from the start of the third cirrus cycle in the second case (Fig. 3b), where KM21_GCM initially predicts a higher ICNC than ML20. For the first case, this is due to the fact that KM21_GCM considers that some portion of the new INP concentration comprises leftover INPs that did not have a chance to nucleate ice by the end of the previous cirrus cycle. As the INP concentration decreases between each cycle, it is assumed that the available INPs are made up of only those that were previously available, meaning no new INPs enter the system. This emulates removal processes in the GCM (e.g. by vertical diffusion or precipitation scavenging). Therefore, our box model calculates ice formation only following the differential AF approach ($\psi$) in Equation 1. As the $S_i$ increases from roughly 1.28 from the end of the first cirrus cycle to 1.3 at the start of the second cirrus cycle, there is a large increase initially in the predicted ICNC. The rate of change of the predicted ICNC for our KM21_GCM approach decreases in subsequent sub-timesteps of the cirrus model as the $S_i$ increases only incrementally. For ML20, despite a larger AF at the start of the second cirrus cycle, the number of newly formed ice crystals that could nucleate onto the fewer number of available INPs does not exceed the pre-existing ICNC. Therefore, no new ice formation occurs until the $S_i$ increases sufficiently after nearly $6\,\mathrm{minutes}$. The error between the two approaches in this case grows in the third cirrus cycle as no new ice formation occurs according to ML20 due to a lower availability of INPs, whereas KM21_GCM predicts higher ICNC due to the increasing $S_i$ and the availability of leftover INPs. The box model works in much the same way for the second case presented in Fig. 3b. The exception is that for the second cirrus cycle there is no new ice formation predicted by KM21_GCM or ML20. The large-scale $S_i$ decreases drastically at the start of this cycle and never grows sufficiently to produce an AF following Möhler et al. (2006) that exceeds the maximum AF from the first cirrus cycle (for KM21_GCM), or produce enough ice to exceed the pre-existing concentration (for ML20). At the start of the third cirrus cycle, the predicted ICNC for both KM21_GCM and ML20 follows the same behavior as the previous example (Fig. 3a).



We also find relatively large non-zero errors between KM21_GCM and ML20 for the cases with decreasing and constant large-scale $S_i$ trends and an intermediate drop in INP concentration (Fig. 2). The profiles of the predicted ICNC and $S_i$ for both of these cases are presented in Fig. 4. We find similar behavior for both cases, with new ice formation predicted in the first cirrus cycle and no new ice formation predicted in the second cycle due to the lower availability of INPs. Non-zero error

for both cases occurs only in the third cirrus cycle after a large increase in the INP concentration ($1000\,\mathrm{L^{-1}}$ to $6000\,\mathrm{L^{-1}}$ for both dust modes). Following KM21_GCM, it is assumed that a significant fraction of the larger INP concentration consists of particles that are new to the system (e.g. from transport with the wind into a theoretical gridbox) and a smaller fraction that did not yet have a chance to nucleate ice. Therefore the model calculates the weighted mean of the new INPs that nucleate ice cumulatively ($\phi$) and the leftover INPs that follow the differential AF ($\psi$) following Equation 3. However, this occurs only in

the second sub-timestep after the $S_i$ increases above 1.2 within the updraft. To emulate the procedure in the GCM, ice formation is not calculated when the large-scale $S_i$ = 1.2 as the Möhler et al. (2006) AF would be zero. Therefore, with our KM21_GCM approach we consider the different ice nucleation behaviors of the available INPs only in the first instance that ice formation can occur. In both cases new ice formation onto the newly available INPs is small as the $S_i$ is relatively low. There is no new ice formation onto any leftover INPs following the differential AF approach because the $S_i$ decreases significantly relative to

the maximum achieved in the previous cirrus cycle.

The maximum (threshold) AF is also recalculated during the first instance of ice formation in the third cirrus cycle for both cases presented in Fig. 4 to account for the larger availability of INPs. This new value is used for subsequent sub-timesteps. For the case with a decreasing trend in large-scale $S_i$ (Fig. 4a), this eventually leads to a large amount of new ice formation according to $\psi$ (Equation 1) as the $S_i$ increases incrementally within the updraft. Meanwhile, ML20 does not predict any

new ice formation during the third cirrus cycle in this case due to the relatively low $S_i$-dependent AFs that produce new ice formation that cannot exceed the pre-existing ICNC. This specific case arguably shows the most notable difference between KM21_GCM and ML20. While both approaches account for the number of INPs contained within ice crystals, with scaling the available INP concentration for KM21_GCM and by taking away the pre-existing ICNC for ML20, we find that applying each method in the cirrus sub-model can lead to large differences in the predicted ICNC, which could have implications on

cirrus climate impacts.

For the case with a constant trend in large-scale $S_i$ (Fig. 4b), the recalculated maximum AF following KM21_GCM leads to new ice formation earlier during the third cirrus cycle than ML20.

Two cases with exceptions for non-zero errors were briefly discussed above (Fig. 2): one for matching large-scale $S_i$ and INP concentration trends (both with an intermediate drop) and the other for constant INP concentration trend with an intermediate

drop in large-scale $S_i$. As the error between these cases is relatively small, and for brevity within this note, we present the predicted ICNC and $S_i$ profiles in Appendix A. However, in summary, we find that our box model behaves similarly for both cases, albeit with different predicted ICNC values due to different INP concentrations. For both of these cases a small error occurs after a large increase of the INP concentration in the third cirrus cycle, similar to the cases presented in Fig. 4. The number of new ice crystals predicted by ML20 is less than that by KM21_GCM, consistent with the results shown previously.





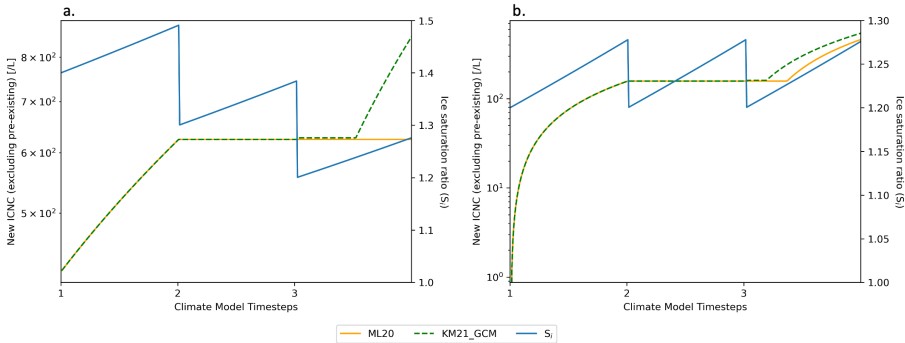

**Figure 4.** Temporal evolution of predicted heterogeneous-only ICNC and $S_i$ for the cases with the intermediate drop in INP concentration for (a) the decreasing large-scale $S_i$ and (b) the constant large-scale $S_i$ across three GCM timesteps. The lines as noted in the legend are consistent with the description under Fig. 3.

This is due to the fact that ML20 considers only the new INP concentration, which forms a sufficient amount of new ice crystals to exceed the pre-existing concentration.

Based on our box model results, the most significant difference between the ML20 and KM21_GCM approaches is the consideration of the previous INP concentration. ML20 mimics the default deterministic, AF-based, approach by Muench and Lohmann (2020) for deposition nucleation on externally mixed mineral dust particles in the ECHAM-HAM GCM. It does not explicitly take the previous INP concentration into account, and instead computes a differential ICNC based on the difference between the number of activated INPs and the pre-existing ICNC. Based on our box model results, it is likely that ML20 under-predicts the number of heterogeneously formed ice crystals under cirrus conditions compared to our KM21_GCM approach as it neglects the different ice nucleation behaviors of available INPs. As the INP concentration changes between GCM timesteps, it is reasonable to assume that some fraction of these INPs is made up of those that were previously present within the model and did not nucleate ice (leftover INPs), and the other fraction is made up of newly available INPs that have not yet been exposed to cirrus formation conditions. The KM21_GCM approach accounts for the leftover INPs and allows them to nucleate ice according to the differential AF, which, if the $S_i$ decreases between GCM timesteps, will be zero. The new approach also allows the new INPs to nucleate ice cumulatively in the first sub-timestep or first instance with suitable ice formation conditions in the cirrus sub-model.

Some of the changes in large-scale $S_i$ and INP concentrations in the cases presented here are extreme and may not occur frequently across timesteps in a GCM, meaning that while we find large errors for some of our cases using the box model, this may not be the case using a GCM. In addition, the KM21 parameterization was developed for a single air parcel within a process model that depicts ice formation within a single cirrus. It does not capture the complexities associated with changes in INP concentrations as well as $S_i$ (among several other factors) across several hundreds of timesteps in a typical GCM simulation. Therefore, we present a short analysis comparing our GCM compatible differential AF parameterization, KM21_GCM, to our default ML20 approach for deterministic heterogeneous ice nucleation in EHCAM-HAM in Section 3.2.



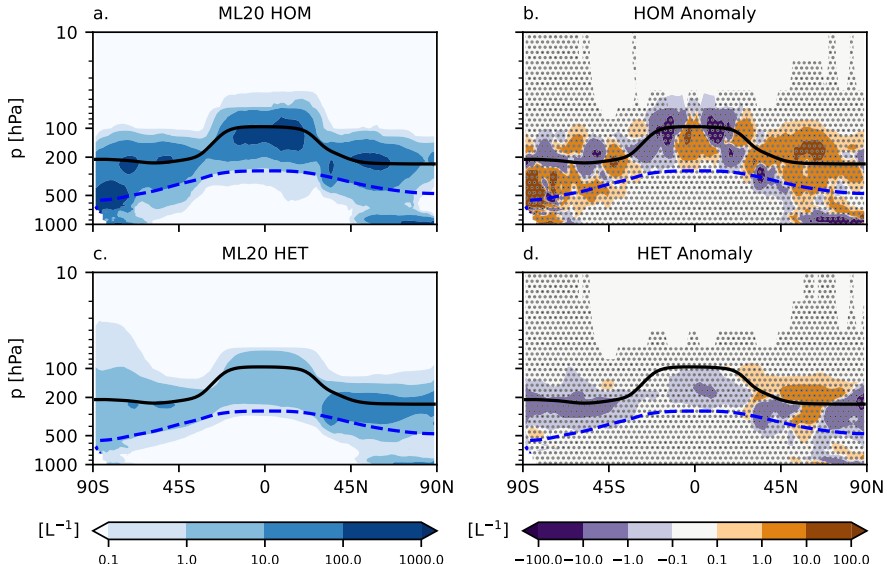

**Figure 5.** Five-year annual zonal mean in-situ cirrus number tracers for ice originating from (a) homogeneous (HOM) and (c) heterogeneous (HET) nucleation for the ML20 simulation. The second column presents the respective ice number tracer anomalies between KM21_GCM and ML20 for (b) HOM and (d) HET. The black line is the WMO-defined tropopause and the blue dashed line is the 238 K contour. The stippling in the anomaly plots shows insignificant data points.

## 3.2   GCM simulations

We conducted two simulations with the ECHAM6.3-HAM2.3 GCM (Stevens et al., 2013; Neubauer et al., 2019; Tegen et al., 2019) to compare the predicted ICNC between the ML20 and KM21_GCM approaches for deterministic heterogeneous ice nucleation within cirrus. Each simulation was run for five years between 2008 and 2012 and follows the same setup as the "Full_D19" simulation by Tully et al. (2022c), which also includes full ice nucleation competition between homogeneous nucleation on liquid sulphate aerosols, immersion freezing on internally mixed mineral dust particles, deposition nucleation on externally mixed mineral dust particles, and vapor deposition onto pre-existing ice crystals.

Fig. 5 presents five-year annual zonal mean ICNC tracers for ice originating from homogeneous (HOM) and heterogeneous (HET) nucleation in the cirrus sub-model (Kärcher et al., 2006; Kuebbeler et al., 2014; Muench and Lohmann, 2020) for ML20, and the HOM and HET anomalies between KM21_GCM and ML20. In this case the HET signal equates to the number tracer for heterogeneously-nucleated ice on mineral dust particles (both internally and externally mixed) as this was the only active INP species for cirrus in our GCM simulations. The stippling in Fig. 5 displays insignificant data points according to the false discovery rate method by Wilks (2016) that accounts for spatial correlation of neighboring grid-points.

In the reference case, ML20, HOM clearly produces more ice in cirrus than HET, consistent with findings by Tully et al. (2022c). The HOM anomaly in Fig. 5b shows a mixed zonal signal by at most $\pm\,100\,\mathrm{L}^{-1}$. This large change by roughly the



same order of magnitude as ML20 HOM indicates that the change in HET parameterizations between ML20 and KM21_GCM affects ice nucleation competition within cirrus. One would expect that areas of positive HOM anomalies would correspond to areas of negative HET anomalies, and vice versa. However, we find this is not necessarily the case in Fig. 5d. It also appears that KM21_GCM produces less HET on average than ML20 in most areas, which is inconsistent with our box model results. For those simulations KM21_GCM predicted higher ICNC values than ML20 (Section 3.1) due to large changes in large-scale $S_i$ or INP concentration across the three GCM timesteps we emulated. As this is not the case in our GCM simulations, it likely means that either such large changes in large-scale $S_i$ and INP concentrations across GCM timesteps do not occur frequently if at all in our model, or that other factors such as temperature and updraft velocity influence our GCM results that we did not test in our box model.

In the tropics, just above the equator, we find that KM21_GCM produces less HET than ML20, which corresponds to an increase in HOM around the same region. Less HET in this area means that $S_i$ growth is unimpeded with the updraft such that high values suitable for HOM occur more readily to produce more ice crystals by this process. This is only partially reflected in zonal profiles of cloud fraction and relative humidity (RH) anomalies in Fig. 6, where there are only small positive anomalies in the southern hemisphere (SH) tropics of up to 1 % that are insignificant (as denoted by the stippling).

There are significant, positive cloud fraction and RH anomalies between 1 and 10 % towards the mid-latitudes and the poles in both hemispheres. In the SH this corresponds to less frequent HET (Fig. 5d), which may allow more frequent high RH values suitable for HOM (Fig. 5b). As HOM occurs as a stochastic process in our cirrus sub-model, the more numerous ice crystals forming in this way contribute to a higher fraction of cirrus. However, the HOM signal is not consistent throughout the SH and is insignificant. There is a much clearer signal in the northern hemisphere (NH) mid-latitudes (roughly 45 °N - 60 °N) where both HOM and HET produce more ice in KM21_GCM than in ML20. For HET this is consistent with our box model results (Section 3.1) that showed that KM21_GCM produced more ice crystals from deterministic ice formation processes. As KM21_GCM accounts for different ice nucleation behavior of available INPs, we found that it often allows for higher rates of ice formation in cirrus. The increase in HOM in the NH in Fig. 5b may be a result of additional latent heat release from more HET that causes air to rise and cool adiabatically, and caused by additional LW cloud-top cooling (Possner et al., 2017) from the higher cloud fractions we find in this area (Fig. 6). However, as HOM and HET, cloud fraction, and RH show positive anomalies, this is likely a systematic signal we find in the model.

The zonal mean HOM and HET ICNC tracer anomalies in Fig. 5 are both insignificant for the five years we tested. Therefore, it is difficult to describe the exact effect of choosing one deterministic ice formation parameterization (ML20 or KM21_GCM) over the other. While there are relatively large, but insignificant changes in ice nucleation competition, the maximum positive and significant anomaly for cloud fraction is 3.6 %. Nevertheless, these changes correspond to only a small positive top-of-atmosphere (TOA) warming effect by around $0.02 \pm 0.35 \, \mathrm{Wm^{-2}}$ that is driven predominately by a weaker shortwave (SW) cloud radiative effect (CRE). Zonally the SW and LW CRE components are insignificant on a 95 % confidence level, except for a small region in the tropics (not shown).

Insignificant differences between ML20 and KM21_GCM in our GCM simulations relative to our box model simulations (Section 3) are not entirely unexpected. In the box model the changes in large-scale $S_i$ and INP concentrations between each





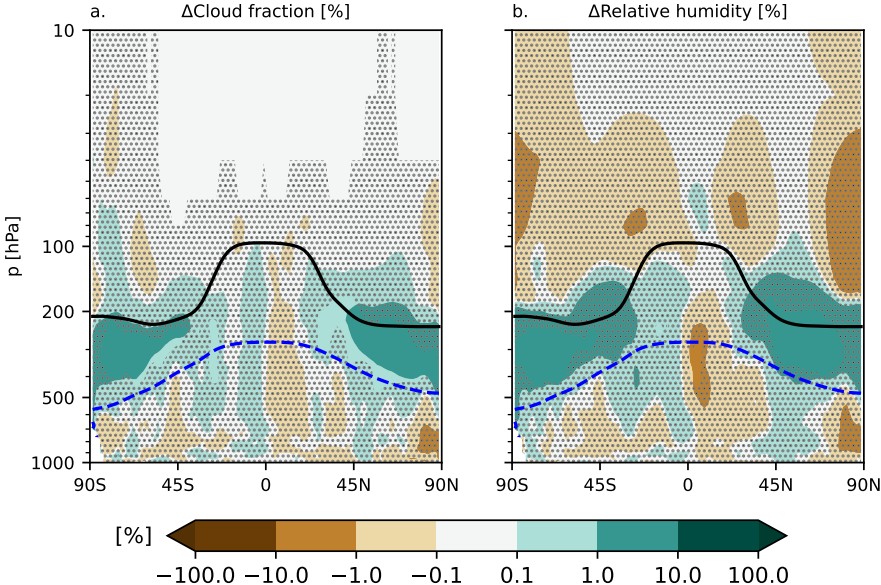

**Figure 6.** Five-year annual zonal mean (a) cloud fraction and (b) relative humidity anomalies between KM21_GCM and ML20. The black line is the WMO-defined tropopause and the blue dashed line is the 238 K contour. The stippling in the anomaly plots shows insignificant data points.

cirrus cycle were extreme in some cases, leading to large differences in the predicted ICNC. Such extreme changes are not unlikely in a GCM, but may occur with a low frequency. In addition, we tested only heterogeneous nucleation in our box model, whereas we included full nucleation competition in our GCM simulations, following Gasparini and Lohmann (2016) and Tully et al. (2022c), that includes vapor deposition onto pre-existing ice crystal. This process was shown to have a large impact on cirrus properties in ECHAM-HAM as it prevents $S_i$ values from rising to high values after a sufficient number of ice crystals already formed (Kuebbeler et al., 2014; Gasparini and Lohmann, 2016).

Our GCM results do show that the choice of deterministic ice formation parameterization (ML20 versus KM21_GCM) has an impact on ice nucleation competition within cirrus. However, the small and insignificant changes in cirrus properties indicate that deposition nucleation onto externally mixed mineral dust particles does not contribute significantly to the total ice number in in-situ cirrus. In fact, Tully et al. (2022c) showed that homogeneous nucleation dominates the ICNC with much higher nucleation rates than all heterogeneous nucleation processes combined (immersion freezing and deposition nucleation, see their Appendix). Therefore, we argue that while the KM21_GCM approach is closer to first principles by accounting for different ice nucleation behavior of available INPs, the implicit budgeting approach by ML20 makes for a much simpler parameterization within a GCM, and it does not require additional tracers. The simulated climate with these two parameterizations is very similar in terms of TOA net radiation, despite significant differences between cirrus cloud fractions.



## 4    Conclusions

In this study we compared two approaches for simulating deterministic heterogeneous ice formation processes: a GCM-compatible version of the differential AF parameterization introduced by Kärcher and Marcolli (2021), KM21_GCM, and

the default approach in the ECHAM-HAM GCM based on cumulative AF by (Muench and Lohmann, 2020), ML20. In a series of simulations using a box model, based on the cirrus sub-model of ECHAM-HAM (Kärcher et al., 2006; Kuebbeler et al., 2014; Muench and Lohmann, 2020), we found that ML20 under-predicts the number of ice particles originating from heterogeneous nucleation relative to KM21_GCM in cases when the large-scale $S_i$ and INP concentration trends differed across the three cirrus cycles we simulated. This is due to the fact that ML20 does not explicitly consider changes in INP concentrations

across GCM timesteps, nor does it consider the different ice nucleation behaviors of available INPs. KM21_GCM takes these factors into account, and allows new INPs to nucleate ice according to the cumulative AF and leftover INPs to nucleate ice according to the differential AF.

The large-scale $S_i$ conditions and the changes in INP concentrations between cirrus cycles that we tested with our box model were rather extreme and may not occur frequently in a GCM. In addition, we conducted only short simplified tests using our box

model that do not capture other changes, namely in temperature and updraft velocity, that would occur across GCM timesteps. As a result, we extended our analysis of ML20 and KM21_GCM with two additional tests with the ECHAM-HAM GCM. We found that choosing between the two deterministic ice formation approaches has an impact on ice nucleation competition within cirrus. However, the signal is mostly insignificant for the five years that we tested (2008-2012), and is inconsistent with the findings from our box model simulations, except in the NH. It is likely that this signal is systematic within the model as

cloud fractions and RH also increase. Overall, the radiative impact of switching between these two approaches is small and insignificant.

While the KM21_GCM approach with explicit INP-budgeting is closer to first principles when simulating deterministic ice formation in an iterative way following the adiabatic ascent of an air parcel, it requires additional tracers in the climate model. Not only does this require additional memory allocation, but it also introduces more room for potential error. It also complicates

the parameterization for determining heterogeneous nucleation on externally mixed mineral dust particles in our cirrus sub-model as we must consider changing conditions across GCM timesteps. Arguments are emerging that call for a simplification of cloud microphysical processes within GCMs, especially in the case that the simplified model is "equifinal" to the more complex version (i.e. the outcome is similar), (Beven, 2006; Proske et al., 2022). ML20 is a simpler parameterization for deterministic ice nucleation than KM21_GCM as it does not require tracing the maximum AF achieved in a cirrus formation cycle or the

INP concentration across GCM timesteps. Given the insignificantly different GCM results between these two approaches, we argue that from the perspective of understanding the impact of cirrus on the climate, the ML20 approach is suitable and more interpretable.

## Appendix A:  Box model simulations of deterministic ice formation



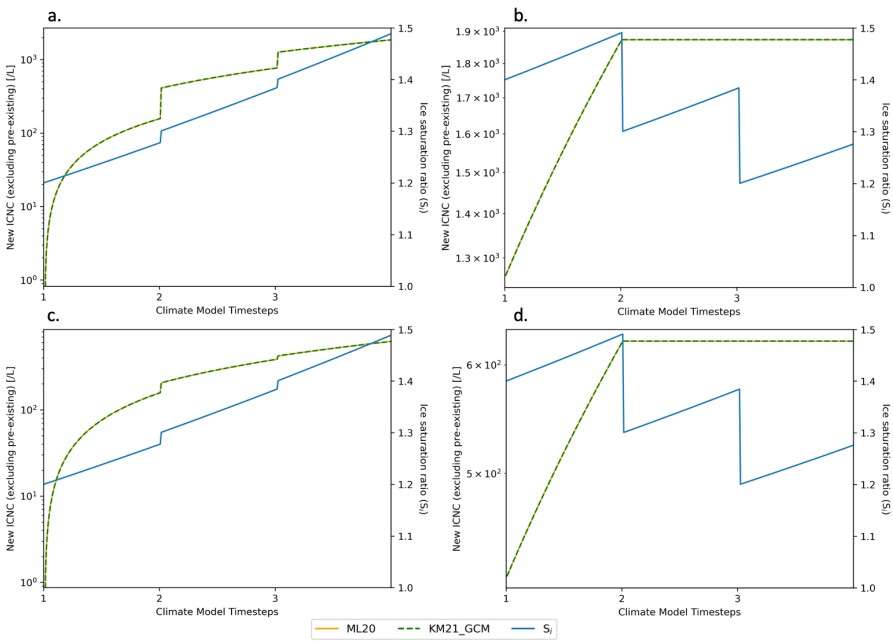

**Figure A1.** Temporal evolution of predicted heterogeneous-only ICNC and $S_{i,seed}$ for four of the six cases that showed agreement between KM21_GCM and ML20 for (a) increasing large-scale $S_{i,seed}$ and INP concentration trends, (b) decreasing large-scale $S_{i,seed}$ and INP concentration trends, and for constant INP concentration with (c) increasing large-scale $S_{i,seed}$ and (d) decreasing large-scale $S_{i,seed}$. The lines as noted in the legend are consistent with the description under Figure 3.

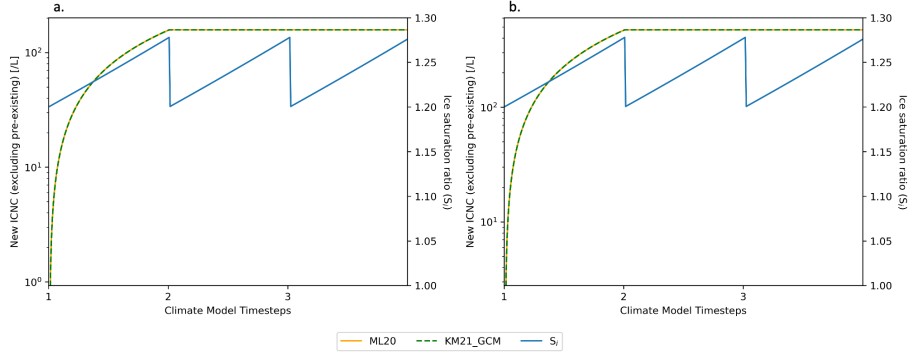

**Figure A2.** Temporal evolution of predicted heterogeneous-only ICNC and $S_{i,seed}$ for two of the six cases that showed agreement between KM21_GCM and ML20 for the constant INP concentration trend with (a) constant large-scale $S_{i,seed}$ and (b) decreasing large-scale $S_{i,seed}$. The lines as noted in the legend are consistent with the description under Figure 3





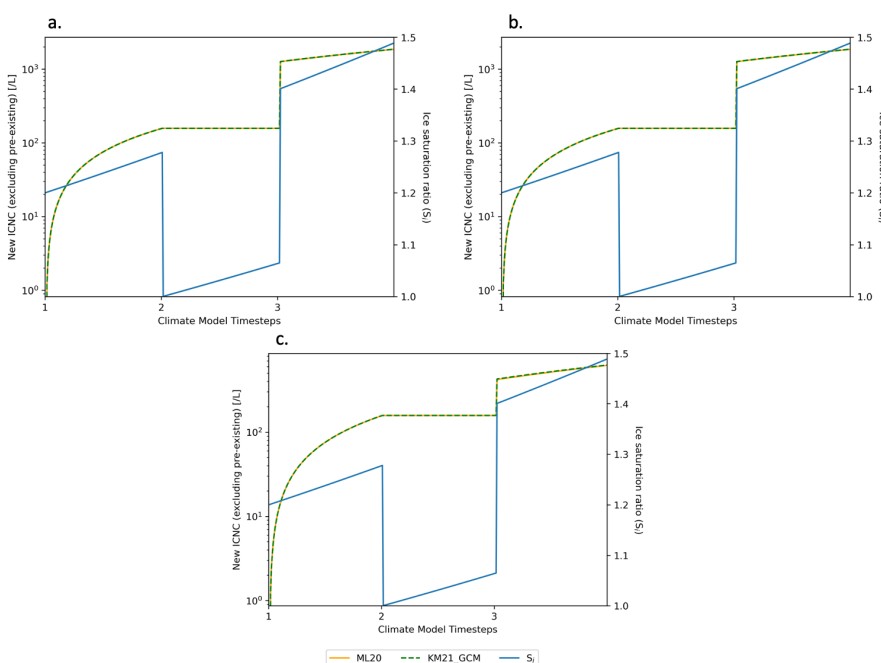

**Figure A3.** Temporal evolution of predicted heterogeneous-only ICNC and $S_{i,seed}$ for three cases with an intermediate drop in large-scale $S_{i,seed}$ that had the smallest non-zero error ($< 1\%$) as shown in Figure 2 for (a) increasing, (b) an intermediate drop, and (c) constant INP concentration. The lines as noted in the legend are consistent with the description under Figure 3.





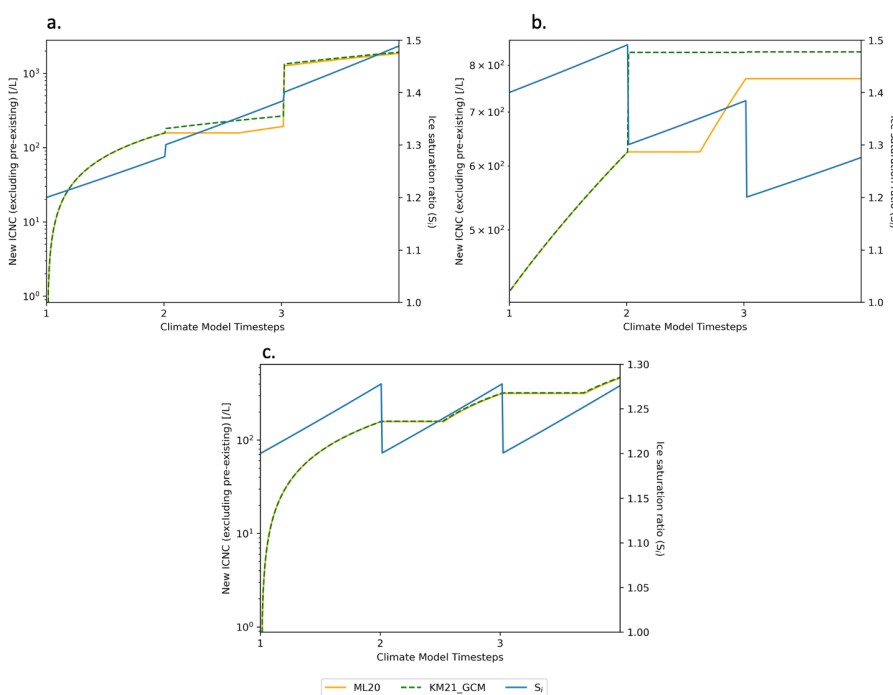

**Figure A4.** Temporal evolution of predicted heterogeneous-only ICNC and $S_{i,seed}$ for (a) increasing large-scale $S_{i,seed}$ and an intermediate drop in INP concentration, (b) decreasing large-scale $S_{i,seed}$ and increasing INP concentration, and (c) constant large-scale $S_{i,seed}$ and increasing INP concentration. The lines as noted in the legend are consistent with the description under Figure 3.



*Code and data availability.* The ECHAM-HAMMOZ model is freely available to the scientific community under the HAMMOZ Software License Agreement, which defines the conditions under which the model can be used (https://redmine.hammoz.ethz.ch/projects/ hammoz/wiki/2_How_to_get_the_sources, last access: 19 October 2022). The specific version of the code used for this study is archived in the ECHAM-HAMMOZ SVN repository at https://svn.iac.ethz.ch/external/echam-hammoz/echam6-hammoz/tags/papers/2022/Tully_et_ al_2022_GMD_for-review (last access: 21 October 2022). More information can be found on the HAMMOZ website (https://redmine. hammoz.ethz.ch/projects/hammoz, (last access: 19 October 2022). The box model that is based on the ECHAM-HAM code that was used to produce the heterogeneous nucleation-only plots in this manuscript, as well as other post-processing and analysis scripts are archived on Zenodo (Tully et al., 2022b). The processed GCM output data to produce the relevant plots in this manuscript are also available on Zenodo (Tully et al., 2022a)

*Author contributions.* CT translated the FORTRAN code of the climate model into Python for the cirrus box model, performed the sensitivity tests between ML20 and KM21_GCM approaches, and wrote the manuscript, DN formulated the GCM-compatible differential active fraction parameterization and provided assistance implementing the differential AF approach into the box model for GCM compatibility. All authors reviewed the manuscript.

*Competing interests.* The authors declare that they have no conflict of interest.

*Acknowledgements.* This Project is funded by the European Union under the Grant Agreement No. 875036 (ACACIA). The GCM simulations were performed on the Euler cluster operated by the High Performance Computing group at ETH Zurich. The authors would like to thank Nadia Shardt for providing tips on improving the discussion of deterministic ice nucleation predictions.



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
