# Peer review of "Technical Note: assessing predicted cirrus ice properties between two deterministic ice formation parameterizations"

_EGUsphere, 2022_

## Referee Comment (RC2)

Review of "**Technical Note: assessing predicted cirrus properties between two deterministic ice formation parameterizations**", submitted to *GMD*.

**General comments.** This technical note describes box model and GCM tests of two approaches for treating deterministic ice nucleation in climate models. The two methods are based on a differential activated fraction (AF) approach (KM21) and the current scheme in the ECHAM model that is a cumulative AF scheme (ML20). The authors find some substantial differences between the two approaches for some cases in the box model, but overall only a very small impact in the GCM. They conclude that the simpler ML20 scheme may be appropriate given the similarity in results compared to the more detailed KM21 scheme.

Overall, this is an important subject and I think the methodology is reasonable. My main criticism is that the writing is confusing in many places, particularly when describing the ice nucleation approaches. Several detailed comments and suggestions related to this are given below in major comments. I also have numerous minor comments and suggestions followed by a handful of editorial comments. Overall, my recommendation is minor revision. However, I want to emphasize that the description of the two ice nucleation approaches and the overall presentation quality need to be improved before I would recommend accepting the paper.

**Recommendation:** *Minor revision*

**Major comments.**

1. As mentioned above, the description of the approaches is rather confusing in several places. In particular, the top part of p. 5 is confusing when discussing the KM21 approach. Some specifics:

 a. It's not clear on p. 5 (though it is explained later in the paper) if the implementation of KM21 here considers previously removed INP (e.g., by including N0 as a prognostic variable following the "explicit INP-budgeting"). This should be explained clearly up front on p. 5.

 b. I would remove the quotes around "differential AL" on line 127, otherwise this makes it confusing what psi actually is.

 c. The example applying Eq. 1 does not seem correct, or at the least it is very confusing. As I understand it here, AF (psi) = 0.05 for both steps. Since 5 L-1 is removed from the total IN in the first step (with "explicit INP-budgeting" in this approach), Eq. 1 should give the correct total of 9.5 L-1 ICNC after two steps (5 L-1 from the first step, plus 4.5 L-1 during the second step). It's stated on line 135 that "phi is based on N0" which then does not include the removal of INPs activated during the first step (that is, no INP-budgeting). Then Eq. 1 gives ~10 L^-1 which is incorrect and too large. But it's not stated here (only later) that for the implementation of this approach the INP are tracked and they are removed from the population when nucleated, that is, that N0 is tracked as a new prognostic variable (or "tracer" as the authors call it). Again, it should be clarified here that N0 is tracked when implementing this approach, so that it does

account explicitly for previously nucleated INPs (related to comment a above). Overall, the example given here on p. 5 gives the impression that the KM21 approach will overestimate ICNC, but I don't think this is not the case when N0 is tracked as an additional tracer.

2. For implementation of the ML20 approach (Eq. 2), are all ice species counted as ICNC (including cloud ice, snow, etc). Or is only cloud ice included?

3. For implementation of both approaches in the box model, there are also many confusing aspects (especially for KM21). Again, some specific issues:

d. Near lines 181-182, it's stated that phi is set to the maximum AF in the cycle, and does not decrease in subsequent cycles. But this raises the question of how INP are recovered (i.e., how phi is relaxed to the background value phi = 0). This is clarified in the next paragraph on line 188 (phi is to 0 outside cloud), but I would state it in the paragraph around lines 182-183 as otherwise this is confusing.

e. For implementation in the GCM, presumably phi as well as INP are tracked as prognostic variables and advected and diffused? Of course, there is no transport in the box model, but it should be mentioned how the tracked phi and INP are treated in the full 3D GCM.

f. Relatedly, also on line 181, in the box model it seems that phi is not actually added as a tracer (prognostic variable), but rather $ICNC_{i-1,j=n}$ (see line 203). Please be very clear about which exact variables are actually tracked in the implementation of these schemes.

g. p. 6, eq. 3. This equation is confusing. Why are i and j different indices? Are both time indices? How are they different? Also, line 192 is confusing. I think by "the previous AF" you mean the maximum AF from the previous cycle? If so, please clarify this and use more precise wording. It's also not clear why $phi_{i-1,j=n}$ has 2 subscripts, when all previous instances of phi have a single subscript. Why? Again, what do i and j actually indicate, and how are they different?

4. To improve the presentation and flow, I suggest describing the default ML20 approach before the newer KM21 approach. Thus, switch sections 2.1.1 and 2.1.2?

5. A key limitation of the box model is lack of sedimentation. Around lines 320-322 you mention that some of the large-scale tendencies for Si and INP concentrations in the box model tests may not be realistic, but it seems the lack of ice transport and sedimentation may be a bigger issue when comparing the box model and GCM results. Also see lines 347-350 where it's implied that the biggest differences between the box model and GCM are the magnitude of changes in Si and INP concentrations, not mentioning the role of precipitation generation and sedimentation in the GCM at all. The sentences on lines 376-377 are also relevant here.

6. Line 162. Where does the downdraft part fit it? Does the model include an updraft-downdraft cycle? Or is it only updraft. More description is needed here – this was very confusing.

7. Lines 220-225. It seems that each cirrus cycle is assumed to have a timescale equal to the GCM time step. This seems like a major assumption and difficult to justify on physical grounds. What is the sensitivity to this? That is, what if there was more than 1 cycle per GCM time step, or less than 1?

8. You use term "error" when describing differences between the two approaches. Since there is no benchmark or truth, I feel "difference" would be a better term to use than "error".

**Minor comments.**

1. Abstract. Could you explain the differential activated fraction (AF) approach in a simple way while staying within the abstract length limit (e.g., simplify the description on lines 72-75 for here)? This would be useful for readers who aren't familiar with this approach so they can more fully understand the abstract.

2. Line 72. Would it be clearer to say that the differential AF approach describes the number of INPs that are active *per temperature interval*, rather than "within a certain temperature interval" as currently written. Taken literally, the latter does not necessarily imply a differential quantity (i.e., units of IN number per temperature).

3. Lines 80-82. I think the term "INP-budgeting" should be "explicit INP-budgeting", in contrast to the "implicit INP-budgeting" in the ML20 approach as mentioned on line in the abstract. Also, not clear why explicit INP budgeting might over-predict ICNC. Can this be explained better?

4. Line 90-91. Can this be explained more? What is the differential ICNC approach, and what is the issue stated by Karcher and Marcolli (2021)?

5. Lines 98-99. But according to line 14 in the abstract, GCM simulations were also run and analyzed as part of this study. I'd mention that here.

6. Lines 115-119. These two sentences seem to contradict one another. Perhaps in the first sentence, reword to "such that all of the available aerosols that can potentially serve as INPs nucleate ice during a single substep of the cirrus sub-model".

7. Line 135. I would replace "should be" with "will be", as "should be" could be mistaken to imply what the concentration would be if there were no error.

8. Line 140. It's not the values per se that are relevant here, but the magnitude of error. Thus, I'd suggest replacing "values" with "error" and then "are not large" with "is not large" for subject-verb agreement.

9. Line 186. The "tracers" are tracked in time (and time and space in the GCM), so I think it's confusing to say you added a tracer for the "previous INP concentration ($N\_0,i-1$)". Better to say you added a tracer for the INP concentration (N0).

10. Lines 230-234. The description in these sentences is very confusing. The way this is written makes it seem like most cases have small error (< 1%), but this is true for only 9 of the 16 cases. Please rewrite this to make it clearer.

11. Line 237. It's confusing to say that ML20 only considers the starting INP concentration, because the INP concentration changes over time in most runs (e.g., Table 1). Or do you mean it only considers the initial IN concentration at the start of each cycle? If so, please clarify this.

12. Figure 3. Perhaps label which cases a) and b) are at the top of the plots, rather than only mentioning this in the figure caption.

13. Line 298. Not clear what you mean by "with exceptions for non-zero errors". Can you reword this?

14. Line 300. I don't think you mean "error between these cases", but rather the "error for these cases".

15. Same comment for Fig. 4 as Fig. 3 above.

16. Lines 336, 341, and elsewhere. What do you mean by the HOM and HET anomalies? Does this just mean the difference in HOM and HET between KM21_GCM and ML20? In which case, I'd replace "anomaly" with "difference" and "anomalies" with "differences" throughout.

17. Appendix. Similar comment for all the figures here as my previous comments for Fig. 3-4.

**Editorial/technical comments.**

Line 24. I'm not sure what the convention for GMD is, but should acronyms be defined again in the main text even if they're defined first in the abstract? I would lean toward yes.

Line 29. Remove the second comma.

Line 44. "that is sparsely populated" seems somewhat awkward wording. Perhaps something like "that has low concentrations"? Similarly, perhaps on line 51 replace "are also sparsely populated" with "are sparse".

Line 69. Again, I would define AF here (first time it's used in the main text).

Line 77. Suggest replacing "that can occur" with "occurring".

Line 84. Add "approach" after "AF"?

Line 85. "This method demonstrated that it is able to counteract" is confusing. I think you mean "Karcher and Marcolli (2021) demonstrated that this method is able to counteract…"

Line 125. Add comma before "when".

Line 128. Change the second "of" to "to" or "in".

Line 269. I think "produce" should be "produces"?

Line 308. Remove comma right before "approach".

Figure A2 caption. Period is missing at the end of the last sentence.

---

## Author Response (AR1)

**Technical Note: assessing predicted cirrus ice properties between two deterministic ice formation parameterizations (egusphere-2022-1057)**

*Colin Tully, David Neubauer, and Ulrike Lohmann*

**Referee #1 Author Response**

To Referee #1,

Thank you for taking the time to review our manuscript and for providing useful comments on improving this study. We have quoted each of your general and specific comments below with our response and changes in the text where applicable. We omitted minor changes like typos and word removals. In some cases, our responses were linked to more than one of your comments, which we note below.

Sincerely,
Colin Tully (on behalf of all co-authors)

**General Comments**
1. **Comment:** There are a few contradictions in this manuscript that need to be addressed. Kärcher and Marcolli (2021) motivated the differential AF approach by arguing that the cumulative AF approach may overpredict heterogenous ice nucleation, yet the box model results featured here show that when ICNC does not agree, the cumulative AF approach underestimates ice nucleation compared to differential AF. The box model results also show that predicted ICNC between the two schemes frequently agree (and at worst show a discrepancy of < 2X in extreme conditions, looking at Fig. 2), yet an argument is made later in the manuscript when explaining impacts of KM21 on cirrus simulated in ECHAM-HAM that this ice nucleation parameterization often increased ICNC compared to ML20. Also, though KM21 sometimes increased ICNC in the box model results, it decreased zonal mean heterogeneously nucleated ICNC compared to ML20 over much of the SH and NH. Perhaps these issues could be resolved with more careful discussion of agreement and significance. Otherwise, I suggest addressing each of the contradictions explicitly in the discussion.
   a. **Response:** KM21 refers to the parameterization as presented by Kärcher and Marcolli (2021). We formulated a GCM-compatible version (KM21_GCM) to compare to our default approach in ECHAM-HAM (ML20). For greater clarity we replaced the schematic in Figure 1 with a video supplement, in line with your **Comment 21** under the **Specific Comments** section, that explains the differences between KM21, KM21_GCM, and ML20. In summary, KM21 addresses the issue that one should not use cumulative AF with INP-budgeting as it could overestimate the number of newly formed ice crystals. ML20 follows a

cumulative AF approach, but does **not** explicitly budget INPs, instead using a differential ICNC variable. In theory these two approaches should lead to similar results. However, this is only applicable for a single cycle of cirrus ice formation. In a GCM cirrus ice formation is calculated at every timestep, of which there could be several thousand during a typical simulation. In addition, INP concentrations and ice saturation ratio conditions can change across GCM timesteps, among other factors. Therefore, we formulated KM21_GCM to account for these changes across GCM timesteps and to consider the "different ice nucleation behaviors of available INPs". This is discussed in more detail in the supplementary video.

b. Regarding the contradictions that you cite in the predicted ICNC, this is related to the formulation of KM21_GCM, the limitations with the box model relative to the GCM, and the rather extreme conditions we tested in the box model. These are either directly addressed or are related to issues you pointed out under **Comments 15, 16, 21, 28, 34, 35, 41, 45 and 46** in the **Specific Comments** section.

2. **Comment:** The author also argue in the abstract and conclusion that ML20 leads to increased interpretability of GCM results, but I think more discussion is needed to support this argument. The formulation of ML20 is indeed simpler, but explicit INP budgeting makes more sense intuitively than the implicit treatment of INP removal in ML20 (e.g., ..."ICNC is updated only if the amount of new ice formation as a portion of N0 exceeds ICNC from the previous sub-timestep"). In short, an argument for increased interpretability could be made for either scheme, so I don't see this as a clear benefit of ML20, though there are other obvious benefits that are described in the conclusion.

   a. **Response:** We agree that arguments can be made on the interpretability of the results from either approach. The wording that we chose was incorrect in this case. We edited the text in the abstract (see under **Comment 1** in the **Specific Comments** section) and in the conclusions to reflect this correction under **Comment 47** in the **Specific Comments** section.

   b. **Changes in the text:**

*"While the KM21_GCM approach with explicit INP-budgeting is closer to first principles when simulating deterministic ice formation in an iterative way following the adiabatic ascent of an air parcel, it requires additional tracers in the climate model. Not only does this require additional memory allocation, and thus greater CPU demand, but it also complicates the parameterization for determining heterogeneous nucleation on externally mixed mineral dust particles in our cirrus sub-model as we must consider changing conditions across GCM timesteps, which also means that there is increased likelihood of unintended errors within the model code. Arguments are emerging that call for a simplification of cloud microphysical processes within GCMs, especially in the case that the simplified model is "equifinal" to the more complex version (i.e. the outcome is similar), (Beven, 2006; Proske et al., 2022). ML20 is a simpler parameterization for deterministic ice nucleation than KM21_GCM as it does not require tracing the maximum AF achieved in a cirrus formation cycle or the INP concentration across GCM timesteps. As our results showed that differences in cloud*

*properties as well as radiative effects were insignificant, we argue that from the perspective of understanding the impact of cirrus on the climate, the simpler ML20 approach is suitable."*

3. **Comment:** Please also check that all tables and figures appear in the section in which they are first referenced.
   a. **Response:** This is an easy LaTeX fix that we implemented in the revised manuscript. This will, of course, be addressed for eventual typesetting.

**Specific Comments**
**Abstract**

1. **Comment:** I think it is important to point out in the abstract that this study focuses on deposition mode ice nucleation only, possibly in the title, and to allow for the possibility that explicit budgeting could still impact cirrus properties through immersion mode nucleation. As you state on L64, immersion mode is likely the dominant mode of heterogenous ice nucleation in cirrus. It would also help the reader to briefly describe the difference between explicit and implicit INP budgeting.
   a. **Response:** These are good points that we neglected. We added the reference to deposition nucleation to the abstract instead of to the title. We also reformulated the middle section of the abstract to differentiate between the explicit and implicit INP-budgeting approaches more clearly. Finally, we added a reference to the fact that this study could be extended to assess the impact our new approach would have on immersion freezing. We also updated Line 64 as new evidence from Froyd et al. (2022) points to deposition nucleation of mineral dust particles as the most abundant source of ice in the upper troposphere, though in their model they excluded immersion freezing. Please find these changes under your **Comment 11**.
   b. **As you included a few comments on the abstract, we rewrote quite a bit of it.** Therefore, we quote the entire rewritten abstract here for simplicity and refer to these changes in subsequent comments.
   c. **Changes in the text:**

*"Determining the dominant ice formation mechanism in cirrus is still an open research question that impacts the ability to assess the climate impact of these clouds in numerical models. Homogeneous nucleation is generally well understood. More uncertainty surrounds heterogeneous nucleation due to a weaker understanding of the complex physio-chemical properties (e.g. ice nucleation efficiency and atmospheric abundance) of ice nucleating particles (INPs). This hampers efforts to simulate their interactions with cirrus, which is crucial in order to assess the effect these clouds have on the climate system. Kärcher and Marcolli (2021) introduced a new deterministic heterogeneous ice nucleation parameterization based on the differential activated fraction (AF), which describes the number of INPs that activate ice within a specified temperature or ice saturation ratio interval. They argued that this new approach with explicit INP-budgeting, which removes INPs from the total population after they nucleate ice, could help to correct a potential over-prediction of the importance of heterogeneous nucleation within cirrus when budgeting is not considered. We formulated a general circulation model (GCM)-compatible version of the*

*differential AF parameterization for simulating only deposition nucleation within in-situ cirrus and compared it to the method currently employed in the ECHAM6.3-HAM2.3 GCM that is based on cumulative AF. This default cumulative AF approach does not use explicit INP-budgeting, but instead implicitly budgets for INPs that nucleated ice using a differential ice crystal number concentration variable to calculate whether new ice formation should be added to the pre-existing concentration. In a series of box model simulations that were based on the cirrus sub-model from ECHAM, we found that the cumulative approach likely under-predicts heterogeneous nucleation in cirrus as it does not account for interstitial INPs remaining from the previous GCM timestep. However, as the cases that we simulated in the box model were rather extreme, we extended our analysis to compare the differential and cumulative AF approaches in two simulations in ECHAM-HAM. We find that choosing between these two approaches impacts ice nucleation competition within cirrus in our model. However, based on our five-year simulations, the small and insignificant difference in the top-of-atmosphere radiative balance of $0.02 \pm 0.35$ Wm$^{-2}$ means that the overall climate impact is negligible. We argue that while our GCM-compatible differential AF parameterization is closer to first principles, the default approach based on cumulative AF is simpler due to the lack of additional tracers required. Finally, our new approach could be extended to assess the impact of explicit versus implicit INP-budgeting on the ice crystal number concentration produced by immersion freezing of mineral dust particles as this is also an important mechanism in cirrus.*

2. **Comment:** L3: "There is more uncertainty surrounding heterogeneous nucleation processes due to the complex physio-chemical properties of ice nucleating particles." This is a bit of a general, catch-all statement. The authors could consider a different approach to motivating this study. For example, why is it important to improve representations of ice formation in cirrus? Or, if you want to discuss the complexities of INP properties in the abstract, you should elaborate on how their complex properties (by which I assume you mean the myriad types, ice activation mechanisms/chemistry) relate to poor predictive understanding of INP, and how the poor predictive understanding relates to the challenges of representing ice formation processes in cirrus. In other words, you need to hold the reader's hand a bit more to help them understand the motivation for studying two different deterministic INP parameterizations.

   a. **Response:** This is tricky. While we agree that more context is needed, the abstract should be kept as brief as possible, and more detail can be provided in the main text. Therefore, we revised the statement here to be a bit more descriptive and provided more detail in the introduction, which coincides with your **Comment 11** (see changes in the text there). Please see the revised wording for the abstract under **Comment 1**.

3. **Comment:** L4: "...follows a time-dependent (stochastic) or time-independent (deterministic) approach...". This sounds like a point that would be more appropriate in the introduction. I don't think this is a necessary distinction to describe in the abstract because the two parameterizations you are comparing are both deterministic.

   a. **Response:** We agree, and this was excluded from the abstract in favour of more detailed explanations in line with your other comments.

We added this detail in the introduction in the part of the text that corresponds to your **Comment 11**, see changes in the text there.

4. **Comment:** L6: Please define "differential activated fraction"
   a. **Response:** Agreed. This was added to the text and is quoted in our response to your **Comment 1**.

5. **Comment:** L12: "...as it does not account for INP fluctuations across GCM timesteps.". After reading the full draft, I return to this line and understand what you mean here, but I think this phrasing is misleading. Looking at Fig. 1 for example, there are "fluctuations" in INP concentrations at each timestep in both schemes. Perhaps "...as it does not account for interstitial INPs remaining from the previous time step"?
   a. **Response:** Yes, we can see how "fluctuations" is misleading and we agree that more specificity is needed here. We like your wording, so this was amended in the text. See changes in the text under **Comment 1**.

6. **Comment:** L15: "...small and insignificant". Please clarify what is meant quantitatively by small and insignificant.
   a. **Response:** We agree this can be clearer, therefore we added the TOA radiative anomaly in the text to make this distinction. Please see changes in the text under **Comment 1**.

**Introduction**
7. **Comment:** L29: typo at "Climate Change"
   a. **Response:** Thank you for pointing that out. It is fixed in the manuscript, and you will see it in the tracked changes PDF.

8. **Comment:** L31: "..are mostly well established.", "mostly" is unnecessary.
   a. **Response:** We agree, and this was removed from the sentence.

9. **Comment:** L47: "...due to the presence of an INP surface." Please delete "surface".
   a. **Response:** We also agree here, and this was removed as well.

10. **Comment:** L59: "...e.g., on mineral dust (Murray et al. 2021) or on black carbon particles...". Please delete these two instances of "on".
   a. **Response:** This is in line with comment 9 above. These were removed.

11. **Comment:** L65: "In general, as there are more factors that govern the complexities of heterogenous nucleation than homogenous nucleation...". This is another catch-all statement. Again here, please elaborate on how the poor predictive understanding of INP relates to the challenges of representing ice formation processes in cirrus. Also, please define "ice nucleation competition."
    a. **Response:** We argue that the previous paragraph discusses the issues related to heterogeneous nucleation, but after re-reading a clearer link could be made in the text. We also incorporated changes in line with your **Comment 1**.
    b. **Changes in the text:**

*"The theory behind homogeneous nucleation is relatively well understood (Koop et al., 2000; Ickes et al., 2015), with new evidence perhaps suggesting higher freezing onsets at cold temperatures for sulphuric acid droplets (Schneider et al., 2021). However, heterogeneous nucleation in general is still a topic of substantial research (Cziczo and Froyd, 2014; Kanji et al.,2017). Specifically, the ability of certain materials to act as an INP, e.g. mineral dust (Murray et al., 2012), which is likely the most abundant INP species in the atmosphere especially downstream of source regions (Froyd et al., 2022), or black carbon particles (Mahrt et al., 2018, 2020), as well as the characterization of their abundance in the atmosphere (Li et al., 2022). Furthermore, heterogeneous nucleation can occur via several mechanisms. For example, from immersion freezing within a solution droplet or by the deposition of water vapor onto the surface of an INP (Vali et al., 2015; Kanji et al., 2017; Heymsfield et al., 2017), the former of which is thought to be the most common heterogeneous nucleation mechanism in cirrus (Kärcher and Lohmann, 2003), though newer evidence points to the abundance of deposition nucleation in the upper troposphere (Froyd et al., 2022).*

*Generally, the factors discussed above lead to an overall poor predictability of how INPs influence heterogeneous nucleation mechanisms in cirrus and contribute to uncertainties when simulating these mechanisms in numerical models. This makes it difficult to simulate the impact on ice nucleation competition in cirrus, which influences the ability to reliably estimate the radiative effects of these clouds.*

*Due to their coarse resolution, general circulation models (GCMs) rely on parameterizations of both homogeneous and heterogeneous nucleation based on laboratory and field-based measurements of ice formation. These parameterizations can follow either a stochastic (time-dependent) approach based on ice nucleation rates or a deterministic (time-independent) approach. For example, homogeneous nucleation of aqueous solutions droplets is simulated in the ECHAM-HAM GCM following the stochastic approach by Koop et al. (2000) that is based on simplified assumptions of classical nucleation theory. A common method for simulating deterministic ice nucleation mechanisms is based on the activated fraction (AF, or frozen fraction), i.e. the number of ice-active particles at specific temperature and/or ice saturation conditions out of a population of particles (Vali, 1971; Vali et al., 2015; Vali, 2019; Kärcher and Marcolli, 2021). …"*

12. **Comment:** L69: Please define "AF of available INPs."
    a. **Response:** We are not sure what you mean by this comment. Is it the AF that is confusing and we should redefine it? Or is it "available" that is confusing? We changed the text to more clearly define AF. We also redefined acronyms in the main text (e.g. INPs). The "AF of available

INPs" is described in the new video supplement to this study, see your **Comment 21**.

    b. **Changes in the text:**

*"A common method for simulating deterministic ice nucleation mechanisms is based on the activated fraction (AF, or frozen fraction), i.e. the number of ice-active particles at specific temperature and/or ice saturation conditions out of a population of particles (Vali, 1971; Vali et al., 2015; Vali, 2019; Kärcher and Marcolli, 2021)."*

13. **Comment:** L68: "...that are based on laboratory measurements of ice formation.". There are several parameterizations of heterogenous ice nucleation that are derived from field measurements.
    a. **Response:** This is also a good point. We changed the wording in the text to include both aspects of observational-based measurements.
    b. **Changes in the text:**

*"Due to their coarse resolution, general circulation models (GCMs) rely on parameterizations of both homogeneous and heterogeneous nucleation based on laboratory and field-based measurements of ice formation."*

14. **Comment:** L75: "As Vali (1971) explains, on the one hand, the former approach is useful to describe the freezing behavior of a single particle.". Single particle type? Single INP species? Please rephrase or explain how differential AF approach can also be useful for simulating an INP population.
    a. **Response:** After re-reading this statement and the Vali 1971 paper, as well as conferring with group members who conduct laboratory experiments, we concluded that it is not needed in the context of this study, nor is it correct. The two approaches theoretically are connected by taking the integral of the differential spectra to obtain the cumulative. The way this statement is written it makes it sound like they are disparately different. Therefore, we cut this statement and the following statement starting with "On the other hand…" from the revised text.

15. **Comment:** L80: "For example, if a model explicitly removes INPs from the total available population after each ice formation event and adds them to nucleating ice number concentrations...". Please rephrase. Do you just mean that INPs are effectively removed from the population when they trigger ice formation?
    a. **Response:** Yes, this is our explanation of INP-budgeting, but we agree this can be clearer as per your comments above. This statement was shortened in the text to make this clearer.
    b. **Changes in the text:**

*"However, as Kärcher and Marcolli (2021) highlight, care must be taken when determining which approach to use when calculating the number of ice crystals that can form on INPs. This is especially true for numerical models that simulate the temporal evolution of the ice saturation ratio, based on temperature, to calculate new ice crystal formation, like in ECHAM-HAM (Section 2.1). For example, if a model budgets INPs from the total population*

*after they nucleate ice, then using the cumulative AF approach may overpredict the number of heterogeneously nucleated ice crystals as it is based on the total number of INPs that could activate between the freezing onset temperature and a given temperature (Vali 1971, 2019)."*

16. **Comment:** L82: Please elaborate. It is not immediately clear to the reader how the cumulative AF approach would lead to the overprediction of heterogeneously nucleated ice.
    a. **Response:** Yes, we agree here. As this is related to your **Comment 15**, we combined the changes in the text above.

17. **Comment:** L99: Please specify somewhere in this paragraph that this study is limited to deposition mode nucleation.
    a. **Response:** We agree that this should be clear throughout the text, so we added the following statements to the paragraph.
    b. **Changes in the text:**

*"Note, our analysis is applicable only to deposition nucleation mechanisms within in-situ cirrus. Extending the analysis to other ice nucleation mechanisms, namely immersion freezing, is discussed below."*

**Methods**
18. **Comment:** L114: In my opinion, this paragraph should begin with the prior line beginning "Muench and Lohmann (2020)..."
    a. **Response:** After re-reading, we agree, and this was moved to the next paragraph.

19. **Comment:** L124: What is meant by "freezing processes"?
    a. **Response:** Our wording was also incorrect here. For example, deposition nucleation is not a freezing process (liquid to solid), but rather a nucleation mechanism. For clarity, we refer to homogeneous and heterogeneous nucleation as "modes" and the specific ways in which either of these occur as "mechanisms". We found a couple discrepancies in the text that can help resolve this and you will see them in the tracked changes PDF. Along this line, we amended the specific line you quoted in your comment.
    b. **Changes in the text:**

*"Kärcher and Marcolli (2021), hereafter KM21, introduced a new method to describe ice nucleation by the number of activated particles."*

20. **Comment:** L125: What is AF($\phi$)? Is AF a function of ($\phi$)? I see $\phi$ is partially described in L126. Please more explicitly define.
    a. **Response:** No, the AF is not a function of phi. It is the symbol we use for the AF in the equations. As this is unclear, we added a "hereafter" in the brackets.

21. **Comment:** Figure 1: The flow and organization of this figure needs to be improved. Please consider putting the resulting ICNC in a single column and creating a single column for the other features of the blocks, such as "total available INP". You could do the same with subheadings for "Previous INP" and "New INP" for KM21_GCM blocks. The comparable features of each block are not in consistent locations which makes it hard for the reader to follow this figure. Please also specify in the caption whether the INPs in each scenario are interstitial or "total available" or if they are inclusive of previously activated INPs. For KM21_GCM, consider at "+" to indicate that INP from the previous time step are added to the new INP according to AF. Why is the budgeted "leftover" INP in timestep 1 not included in the bottom right blue box for KM21_GCM (i.e., why is $N^*_{i=1,j=n} = 0$?). Please also add the information about AF on L196 to the Fig. 1 caption. A brief explanation about the different INP treatments at $i=2$ in the caption would also help make this figure read more easily.

   a. **Response:** We agree that this figure is rather confusing for a reader and does not capture the complexity of the issues we are examining in this note. Therefore, we decided to cut this figure in the revised manuscript in favour of a video supplement that walks the reader through the differences between KM21, KM21_GCM, and ML20. Included in the video is an explanation of what we mean by the "different ice nucleation behaviours of available INPs", related to your **comments 34 and 40**. We replaced the associated text with the schematic to summarise the content of the video. See the revised text below. The video will be uploaded after acceptance of the manuscript, following GMD submission guidelines.

   b. **Changes in the text:**

*"The video supplement to this study provides more information on the theoretical understanding of our new KM21_GCM approach and compares it to our default ML20 approach. In summary, we classify different ice nucleation behaviors based on the available INP concentration, following Equation 3. In subsequent GCM timesteps, following the first, we assume that some INPs were ice active in the previous GCM timestep, and are thus removed by subtracting the ICNC that formed previously ($ICNC_{i-1,j=n}$) on INPs ($N_{0,i}$). Out of $N_{0,i}$ we assume some fraction is made up of remaining ("leftover") interstitial INPs that did not activate ice in the previous GCM timestep ($N_{0,i-1}$) and the remaining fraction contains particles that are new to the system. To obtain the leftover INP concentration, we also subtract $ICNC_{i-1,j=n}$ from the INP concentration from the previous GCM timestep.*

*In the first instance of ice formation in the current cirrus cycle (current GCM timestep), the leftover INPs nucleate ice according to the differential AF ($\psi_j$) and the newly available INPs nucleate ice according to the cumulative AF ($\phi_j$) as denoted in the numerator of Equation 3. Not only does this approach consider changes in INP concentrations across GCM timesteps, but it also accounts for changes in $S_i$. For example, if in the current GCM timestep the $S_i$ is drastically lower than that in the previous timestep, then no new ice formation will occur on the leftover INPs. However, ice formation can proceed on the newly available INPs if the $S_i$ is sufficient to produce ice according to the AF calculation following Möhler et al. (2006). If the $S_i$ increases in the current cirrus cycle compared to the previous one, then ice formation*

*may occur on both the leftover and newly available INPs. Note that Equation 3 also accounts for decreases in INP concentration across GCM timesteps. In such a case the difference term on the left-hand side of the numerator of Equation 3 is set to zero."*

22. **Comment:** L201: "In order to account for leftover INPs as well as INPs that are still included in ice crystals...". That are removed? Activated and thus removed?
    a. **Response:** As this is related to the specific method of KM21_GCM, we replaced this section in the text with a summary of the new method in the video supplement in line with your **Comment 21**, see above. We also included a summary in the text to accompany the video.

23. **Comment:** L204: "This allows us to properly consider changes...". Remove "properly". Track changes?
    a. **Response:** This text was replaced in the revised manuscript to accompany the video supplement in line with your **Comment 21**, see above.

24. **Comment:** Table 1: Please move to the following section where it is first referenced. I noticed this issue with another figure or two. Please check.
    a. **Response:** Thank you for pointing that out. It's a simple LaTeX fix that you will see in the next version of the PDF. For eventual typesetting, this will of course be addressed.

25. **Comment:** L215: What does "non-exhaustive of the changing conditions..." mean? Please rephrase.
    a. **Response:** It means that the two examples in Figure 1 are not the only two examples of possible INP concentrations and Si values that could be simulated in a GCM. Therefore, they are non-exhaustive. However, as we have replaced Figure 1 with a video supplement, in line with your **Comment 21**, this phrasing is no longer applicable.

**Results**
26. **Comment:** L229: What is "agreement"? Please explain how you consider agreement quantitatively.
    a. **Response:** Good point. We added a quantitative description of this in the text.
    b. **Changes in the text:**

*"Of the 16 tests we conducted, six show agreement between ML20 and KM21_GCM in the predicted ICNC. For these cases we define as a 0% error between KM21_GCM and ML20 for each scenario as denoted by the white shading in Fig. 1."*

27. **Comment:** L233: Please remove "Nevertheless"
    a. **Response:** This was removed.

28. **Comment:** L235: Here, and throughout the paper, there is a contradiction that needs to be addressed, whether here or further down in the discussion (but please reference where that discussion begins here). Given that cumulative AF approaches such as ML20 can overpredict ice nucleation compared to explicit budgeting approaches as you described in the Methods, how is it that KM21_GCM is predicting higher ICNC compared to ML20?
    a. **Response:** You are right that this is unexpected based on the theory presented in the current methods section. In line with your **Comment 21** (in this document) on Figure 1, we edited the methods section to provide more clarity on this matter.
    b. To summarise, there are two issues. Firstly, cumulative AF approaches that use explicit INP-budgeting likely overpredict the number of ice crystals based on the findings by Kärcher and Marcolli (2021). They introduced a new differential AF approach (KM21) to address this specific issue. However, our default approach in the ECHAM-HAM GCM (ML20) uses cumulative AF with implicit INP-budgeting. Finally, KM21 is only applicable in a single formation cycle of a cirrus. In a climate model like ECHAM-HAM this is calculated every model timestep in a sub-model that is called from the cloud microphysics scheme. Furthermore, the INP concentration in the GCM can change between model timesteps. Therefore, we formulated the GCM-compatible version of KM21 (KM21_GCM) to account for interstitial as well as previously activated INPs to compare to the default ML20 approach.

29. **Comment:** L236: "On the one hand, while ML20 considers..." I like this paragraph. The differences between the two treatments are clearly stated.
    a. **Response:** Thank you!

30. **Comment:** L251: "Non-zero error between the predicted ICNC for KM21_GCM and ML20 occurs from the start of the second cirrus cycle in the first case (Fig. 3a), and from the start of the third cirrus cycle in the second case (Fig. 3b), where KM21_GCM initially predicts a higher ICNC than ML20.". Please remove "initially".
    a. **Response:** Agreed as this is applicable throughout the entire second cycle. This was removed.

31. **Comment:** L262: "For ML20, despite a larger AF at the start of the second cirrus cycle, the number of newly formed ice crystals that could nucleate onto the fewer number of available INPs does not exceed the pre-existing ICNC.". The authors present this condition as the main feature of ML20 that causes

the unexpectedly lower ICNC compared to KM21. Please elaborate on the broader implications. Among cumulative AF approaches, is this condition unique to ECHAM-HAM (where ice does not form unless INPs > pre-existing ICNC)? If not, would you expect other cumulative AF schemes to result in increased ICNC as expected according to KM21 and your explanation in the Methods?

    a. **Response:** We are unaware of other cumulative AF schemes that implicitly budget INPs like we do in ECHAM-HAM. For example, the heterogeneous ice nucleation parameterizations by Barahona and Nenes (2009) and Liu and Penner (2005), that are commonly used in the CAM5 GCM, do not explicitly state whether they follow a similar approach as we do. This implicit budgeting feature follows the implementation by Muench and Lohmann (2020), who re-worked the cirrus sub-model code to simplify various parameterizations and improve code readability. Despite this, we edited the text to make this clearer that this is a feature of the implicit budgeting approach.

    b. **Changes in the text:**

*"For ML20, the implicit INP-budgeting approach prevents new ice formation from occurring at the start of the second cirrus cycle, despite a larger AF, as the number of newly formed ice crystals that could nucleate onto the fewer number of available INPs does not exceed the pre-existing ICNC. No new ice formation occurs until the $S_i$ increases sufficiently after nearly 6 minutes."*

32. **Comment:** L279: What is meant by "To emulate the procedure in the GCM..."? Please rephrase.

    a. **Response:** This simply means to copy the method in the GCM code in our box model. We rephrased this sentence.

    b. **Changes in the text:**

*"Following the procedure in the GCM, ice formation is not calculated when the large-scale $S_i$ ≤ 1.2 as the Möhler et al. (2006) AF would be zero."*

33. **Comment:** L291: What is meant by "arguably"? Please elaborate.

    a. **Response:** What was meant by this is that while this case did not show the largest error in the predicted ICNC between ML20 and KM21_GCM, it is notable due to the differences in approaches. For clarity though, we removed "arguably" in the revised text.

34. **Comment:** L313: "Based on our box model results, it is likely that ML20 underpredicts the number of heterogeneously formed ice crystals under cirrus conditions compared to our KM21_GCM approach as it neglects the different ice nucleation behaviors of available INPs.". Again, please discuss the discrepancy between the result, your expectations for overprediction described in Methods and the motivation for KM21. Also, please elaborate on or rephrase "...it neglects the different ice nucleation behaviors of available INPs." I am not sure what this means. Please specify what is "different" and

what entities are being compared. Are you arguing that KM21 better emulates the variability in IN-activity between dust populations, or between individual dust particles? If so, you will need to supplement this section with supporting evidence.

    a. **Response:** This is related to previous comments that you made, specifically **comment 21**. We added a new description and a video supplement to explain the differences between KM21_GCM and ML20 in more detail under comment 21. For this statement, we added a reference to the video supplement.

    b. **Changes in the text:**

*"Based on our box model results, it is likely that ML20 under-predicts the number of heterogeneously formed ice crystals under cirrus conditions compared to our KM21_GCM approach as it neglects the different ice nucleation behaviors of available INPs (refer to the video supplement)."*

35. **Comment:** L322: "...this may not be the case using a GCM." Do you mean it remains to be seen how frequently these errors occur?

    a. **Response:** Yes, and after re-reading this sentence, we can remove the last two clauses as our point was conveyed previously in the same sentence.

36. **Comment:** L336: Please briefly describe the false discovery rate method by Wilks (2016) to help the reader understand the significance testing applied. If this is the only significance test applied, please state that Wilks (2016) is what is referred to throughout the rest of the results and discussion.

    a. **Response:** We agree, and we reformulated and added a couple sentences on this.

    b. **Changes in the text:**

*"The stippling in Fig. 5 displays insignificant data points based on an independent t-test, following the false discovery rate method by Wilks (2016). This approach accounts for high spatial correlation of neighboring grid-points where the null-hypothesis cannot necessarily be rejected. Like Tully et al. (2022c), we calculate a 5% significance based on the inter-annual variability of the five-year simulations. For the remainder of this section, we base significance on this method."*

37. **Comment:** L353: "This is only partially reflected in zonal profiles of cloud fraction and relative humidity (RH) anomalies in Fig. 6, where there are only small positive anomalies in the southern hemisphere (SH) tropics of up to 1 % that are insignificant (as denoted by the stippling).". What is meant by "partially reflected" if the anomalies are insignificant? Also, please be explicit here and elsewhere throughout the manuscript on what is meant by significant.

    a. **Response:** We agree this is unclear. The cloud fraction and RH anomalies in the tropics are small and the fact that they are insignificant, as denoted by the stippling, means that it is unclear whether the increase in HOM we found with KM21_GCM is supported

by these anomalies. We revised the text to reflect this. We also added text above in line with your **Comment 36** on being more descriptive on how we define significance in this section.
   b. **Changes in the text:**

*"It is unclear whether this is reflected in the zonal profiles of cloud fraction and relative humidity (RH) anomalies in Fig. 5. Throughout the tropics we find only small cloud fraction and RH anomalies of around ± 1%; however, these signals are insignificant as denoted by the stippling."*

38. **Comment:** L357: "There are significant, positive cloud fraction and RH anomalies between 1 and 10 % towards the mid-latitudes and the poles in both hemispheres." Please add a reference to Figure 6.
   a. **Response:** Good point. This was added to the text.

39. **Comment:** L358: "However, the HOM signal is not consistent throughout the SH and is insignificant." What is meant by HOM signal?
   a. **Response:** This refers to the anomaly of homogeneously nucleated ice in Figure 5 (now Figure 4). It is defined on Line 354. However, in the revised text we refer to this as "The increase in HOM" to make it clearer.

40. **Comment:** L363: See previous comment on "different ice nucleation behavior of available INPs."
   a. **Response:** This is addressed in the supplemental video in line with your **Comment 21**. The video is now attached to the manuscript that explains what we mean by this phrasing. In summary, we differentiate between "new" INPs and "leftover" (interstitial) INPs as there is no direct communication from the cirrus sub-model back to the aerosol model. Therefore, in subsequent GCM timesteps, we must assume that some of the available INPs will be made up of those that are new to the system and those that are leftover. In our new KM21_GCM approach, we take the ice nucleation behaviour of these leftover particles into account in the differential AF approach.

41. **Comment:** L363: "...we found that it often allows for higher rates of ice formation in cirrus.". Did you track ice formation rates in ECHAM-HAM? Or do you mean KM21 results in higher ICNC? This point about frequency is another contradiction with your previous box model results, in which the results showed that the resulting ICNC between the two schemes frequently agree. The authors further argued that the conditions for which the resulting ICNC differed would occur infrequently in a GCM (L320). Please address this contradiction.

a. **Response:** Yes, we meant that KM21_GCM produces higher ICNC. However, after re-assessing our arguments you are right that our box model does shows that these two approaches often agree, or at least show very little error. This was our oversight and for that reason, we reformulated the wording on Lines 320 and 363. This is in line with your **General Comment 1** and **Comment 45** below.

b. **Changes in the text:**

*"Some of the changes in large-scale $S_i$ and INP concentrations we tested in the box model were rather extreme in order to examine differences between KM21_GCM and ML20. However, our box model setup is limited as we assume a constant temperature and updraft velocity. We also did not consider other processes such as ice sedimentation and mixing that would be simulated in a GCM. Furthermore, the KM21 parameterization was developed for a single air parcel within a process model that depicts ice formation within a single cirrus. It does not capture the complexities associated with changes in INP concentrations as well as $S_i$ (among several other factors) across several hundreds of timesteps in a typical GCM simulation. Therefore, we present a short analysis comparing our GCM compatible differential AF parameterization, KM21_GCM, to our default ML20 approach for deterministic heterogeneous ice nucleation in EHCAM-HAM in Section 3.2."*

*"There is a much clearer signal in the northern hemisphere (NH) mid-latitudes (roughly 45 °N – 60 °N) where both HOM and HET produce more ice in KM21\_GCM than in ML20. While the positive HET ICNC anomaly is consistent with some of our findings from our box model results (Section 3.1) that showed KM21_GCM produced higher ICNC than ML20, it is insignificant for the five years we simulated with the GCM and is only evident in the NH. It is more likely that the GCM results confirm our box model results that show in most cases KM21_GCM and ML20 agree or have a very small error (Fig. 1)."*

42. **Comment:** L370: "While there are relatively large, but insignificant changes...". Please define quantitatively what you consider "relatively large".

a. **Response:** After re-reading this, we agree this is not descriptive. We also see that the anomalies themselves are not necessarily that large relative to the ML20 "reference" case. Therefore, we reformulated this in the text.

b. **Changes in the text:**

*"While the zonal mean HOM and HET ICNC tracer anomalies for KM21_GCM (Fig. 5) are both notable (by at least $\pm$ 10 L$^{-1}$) relative to our reference ML20 simulation, they are insignificant for the five years we tested. Therefore, it is difficult to describe the exact effect of choosing one deterministic ice formation parameterization (ML20 or KM21_GCM) over the other."*

43. **Comment:** L373: "Nevertheless, these changes correspond to only a small positive top-of-atmosphere (TOA) warming effect by around 0.02 ± 0.35 Wm–2 that is driven predominately by a weaker shortwave (SW) cloud radiative effect (CRE).". Is there a reference for the cirrus contribution to CRE in ECHAM-HAM? This would be helpful context.

a. **Response:** We agree that this is a good point for context. Both Gasparini et al. (2016) and Gasparini et al. (2020) quantified the cirrus

CRE in ECHAM. We added these estimates and some context to our findings in the text.

b. **Changes in the text:**

*"Despite this finding, the maximum positive and significant anomaly for cloud fraction is 3.6%, which equates to only a small positive top-of-atmosphere (TOA) warming effect by around $0.02 \pm 0.35$ Wm$^{-2}$ that is driven predominately by a weaker shortwave (SW) cloud radiative effect (CRE). Similarly, the global mean net CRE anomaly between the two cases is indistinguishable from zero, $0.00 \pm 0.32$ Wm$^{-2}$. These radiative anomalies are negligible relative to the estimated CRE from cirrus clouds of 5.7 Wm$^{-2}$ and 4.8 Wm$^{-2}$ by Gasparini and Lohmann (2016) and Gasparini et al. (2020), respectively."*

**Conclusions**

44. **Comment:** L400: "...nor does it consider the different ice nucleation behaviors of available INPs.". Please clarify.

   a. **Response:** This was addressed with the video supplement in line with your **Comment 21** and is summarised in the text in the methods section, see above.

45. **Comment:** L404: "The large-scale Si conditions and the changes in INP concentrations between cirrus cycles that we tested with our box model were rather extreme and may not occur frequently in a GCM.". Would it be possible to calculate the frequency of these conditions from the ECHAM-HAM output?

   a. **Response:** This wording was an oversight on our behalf. The limited box model simulations were merely to understand the differences between the two approaches with large changes in the starting conditions we used as input. However, these tests were limited as they did not consider all of the possible changes that can occur across GCM timesteps. Therefore, we extended the analysis with GCM simulations. We amended the text to reflect this change.

   b. **Changes in the text:**

*"We tested rather extreme changes in the large-scale $S_i$ conditions and INP concentrations between cirrus cycles in our box model to examine the differences between the ML20 and KM21_GCM approaches. However, our setup was limited as it did not capture all of the possible conditions and processes that are simulated in a GCM and that are relevant to assessing cirrus climate effects. Namely, we used a constant temperature and updraft velocity in our box model setup. In addition, we did not consider processes such as ice crystal sedimentation and mixing effects (e.g., entrainment). As a result, we extended our analysis of ML20 and KM21_GCM with two additional tests with the ECHAM-HAM GCM. ..."*

46. **Comment:** L408: "However, the signal is mostly insignificant for the five years that we tested (2008-2012), and is inconsistent with the findings from our box model simulations, except in the NH.". Please define "signal" and "mostly insignificant."

   a. **Response:** Agreed. This is unclear, we reworked this statement to specifically refer to the cirrus ICNC tracer anomalies.

    b. **Changes in the text:**

*"… As a result, we extended our analysis of ML20 and KM21_GCM with two additional tests with the ECHAM-HAM GCM. We found that choosing one of the two deterministic ice formation approaches has an impact on ice nucleation competition within cirrus. However, the cirrus ICNC tracer differences for both homogeneous and heterogeneous nucleation were insignificant between these simulations for the five years that we tested (2008-2012). These results corroborate our findings from our box model simulations, which showed that out of the 16 tests we conducted six showed agreement (0% error) and an additional three tests showed a small error between ML20 and KM21_GCM of 0.3% (Fig. 1). This likely highlights that the GCM was often in similar regimes over the five years of simulation as the tests in our box model that showed zero or small errors.*

47. **Comment:** L414: "Not only does this require additional memory allocation, but it also introduces more room for potential error.". Please elaborate on "room for potential error."
    a. **Response:** Agreed. We reformulated this sentence to combine it with the following one to make this point clearer.
    b. **Changes in the text:**

*"Not only does this require additional memory allocation, and thus greater CPU demand, but it also complicates the parameterization for determining heterogeneous nucleation on externally mixed mineral dust particles in our cirrus sub-model as we must consider changing conditions across GCM timesteps, which also means that there is increased likelihood of unintended errors within the model code."*

**References**
1. Barahona, D. and Nenes, A.: Parameterizing the competition between homogeneous and heterogeneous freezing in cirrus cloud formation – monodisperse ice nuclei, *Atmospheric Chemistry and Physics*, 9, 369–381, https://doi.org/10.5194/acp-9-369-2009, 2009.
2. Froyd, K. D., Yu, P., Schill, G. P., Brock, C. A., Kupc, A., Williamson, C. J., Jensen, E. J., Ray, E., Rosenlof, K. H., Bian, H., Darmenov, A. S., Colarco, P. R., Diskin, G. S., Bui, T., and Murphy, D. M.: Dominant role of mineral dust in cirrus cloud formation revealed by global-scale measurements, *Nature Geoscience*, 15, 177–183, https://doi.org/10.1038/s41561-022-00901-w, 2022.
3. Gasparini, B. and Lohmann, U.: Why cirrus cloud seeding cannot substantially cool the planet, *Journal of Geophysical Research: Atmospheres*, 121, 4877–4893, https://doi.org/10.1002/2015JD024666, 2016.
4. Gasparini, B., McGraw, Z., Storelvmo, T., and Lohmann, U.: To what extent can cirrus cloud seeding counteract global warming?, *Environmental Research Letters*, 15, 054 002, https://doi.org/10.1088/1748-9326/ab71a3, 2020.
5. Liu, X. and Penner, J. E.: Ice nucleation parameterization for global models, Meteorologische Zeitschrift, 14, 499–514, https://doi.org/10.1127/0941-2948/2005/0059, 2005.

**Technical Note: assessing predicted cirrus ice properties between two deterministic ice formation parameterizations (egusphere-2022-1057)**

*Colin Tully, David Neubauer, and Ulrike Lohmann*

**Referee #2 Author Response**

To Referee #2,

Thank you for taking the time to review our manuscript and for providing useful comments on improving this study, especially regarding the description of the two ice nucleation approaches. For greater clarity for readers, we replaced the schematic in Figure 1 with a video supplement that will be made available with the manuscript alongside final publication.

I quoted each of your specific comments below with our response and changes in the text where applicable.

Sincerely,
Colin Tully (on behalf of all co-authors)

**Major Comments**

1. **Comment:** As mentioned above, the description of the approaches is rather confusing in several places. In particular, the top part of p. 5 is confusing when discussing the KM21 approach. Some specifics:
   a. It's not clear on p. 5 (though it is explained later in the paper) if the implementation of KM21 here considers previously removed INP (e.g., by including N0 as a prognostic variable following the "explicit INP-budgeting"). This should be explained clearly up front on p. 5.
      i. **Response:** We added a short statement to make the point that INP budgeting is used with KM21
      ii. **Changes in the text:**

*"Instead, they formulated a differential AF ($\psi$) approach, which considers only the number of particles that can activate as a result of incremental changes in $S_i$ during a timestep j-1 to j. The method is based on the probability that the remaining INPs, following explicit INP-budgeting, in the current timestep j do not become ice-active."*

   b. I would remove the quotes around "differential AL" on line 127, otherwise this makes it confusing what psi actually is.
      i. **Response:** Agreed. These were removed in the revised text.
   c. The example applying Eq. 1 does not seem correct, or at the least it is very confusing. As I understand it here, AF (psi) = 0.05 for both steps. Since 5 L-1 is removed from the total IN in the first step (with "explicit

INP-budgeting" in this approach), Eq. 1 should give the correct total of 9.5 L-1 ICNC after two steps (5 L-1 from the first step, plus 4.5 L-1 during the second step). It's stated on line 135 that "phi is based on N0" which then does not include the removal of INPs activated during the first step (that is, no INP-budgeting). Then Eq. 1 gives ~10 L^-1 which is incorrect and too large. But it's not stated here (only later) that for the implementation of this approach the INP are tracked and they are removed from the population when nucleated, that is, that N0 is tracked as a new prognostic variable (or "tracer" as the authors call it). Again, it should be clarified here that N0 is tracked when implementing this approach, so that it does account explicitly for previously nucleated INPs (related to comment a above). Overall, the example given here on p. 5 gives the impression that the KM21 approach will overestimate ICNC, but I don't think this is not the case when N0 is tracked as an additional tracer.

    i. **Response:** No, psi in the first timestep (j=1) is 0.05 and in the second timestep (j=2) psi = 0.1 (Line 134). The example we provided explains the argument that KM21 were making that the cumulative AF approach should not be used when budgeting INPs, as stated on Line 125 in the original manuscript. We added some short statements in the text in this explanation for clarity.

    ii. **Changes in the text:**

*"As a short conceptual example of their argument (see also KM21 Figure 1), starting from an initial INP population ($N_0$) of 100 $L^{-1}$, the expected ICNC at $\phi_j = 0.1$ is 10 $L^{-1}$. Any approach needs to result in 10 $L^{-1}$ at this AF. However, using the cumulative AF approach as described in KM21, if in the first cirrus model timestep $\phi_{j-1} = 0.05$ under ambient temperature and Si the resulting ICNC after this first step is 5 $L^{-1}$, which equates to $\Delta N = 5$ $L^{-1}$ INPs. With explicit INP budgeting, the resulting population after the first timestep is $N0-\Delta N = 95$ $L^{-1}$. In the second timestep $\phi_{j=2}$ is calculated as 0.1. Using this value alone results in a $\Delta N = 9.5$ $L^{-1}$ and thus a total ICNC after this step of 14.5 $L^{-1}$. This is larger than 10 $L^{-1}$ ICNC, therefore the number of activated particles is over-predicted in this case as the INPs activated in the first timestep were ignored during activation in the second timestep. Using the differential AF approach as presented in Equation 1, with $\phi_j = 0.1$ and $\phi_{j-1} = 0.05$, the resulting $\psi_j$ equates to roughly 0.05. When applying this to the number of available INPs (95 $L^{-1}$), $\Delta N = 5$ $L^{-1}$ bringing the total ICNC after the second timestep instead to 10 $L^{-1}$. Although the resulting error between the ICNC values after the second timestep in this short example is not large, not accounting for previously activated INPs in a correct manner could drastically increase the amount of heterogeneous nucleation on mineral dust particles, leading to vastly different cirrus properties."*

    iii. Regarding your comment on N0, we disagree. This section describes the KM21 method and not how it is implemented in a numerical model where such variables would need to be "traced". We describe this in more detail in Section 2.2 where we describe the cirrus box model.

2. **Comment:** For implementation of the ML20 approach (Eq. 2), are all ice species counted as ICNC (including cloud ice, snow, etc). Or is only cloud ice included?
    a. **Response:** For this study we present only in-cloud ice. The GCM results are based on the P3 ice microphysics scheme, which does not distinguish between ice species. All ice species are counted as ICNC in cirrus, including snow. One snow crystal is then assumed to have a single INP, though several INPs will be in one snow crystal as it is formed of several ice crystals. In this case, by using P3 in the GCM, we underestimate the previously activated ice crystals between GCM timesteps by not accounting for collision and coalescence. However, our GCM results show that ML20 and KM21_GCM do not lead to drastic differences in the model so this does not appear to be an issue for cirrus. This caveat was added to text in the conclusions after the text added for your **Comment 5**.
    b. **Changes in the text:**

*"… An additional caveat to our GCM results is that we used the P3 ice microphysics scheme (Morrison and Milbrandt, 2015; Dietlicher et al., 2018, 2019; Tully et al., 2022c), which does not distinguish between different ice hydrometeors. Cloud ice and snow are both considered to make up the total ICNC in cirrus in our study. In our model it is assumed that a snow crystal includes a single INP, whereas there are numerous INPs associated with snow crystals as they are made of several ice crystals. Therefore, by not considering collision and coalescence processes, our model may underestimate the number of previously formed ice crystals. This is unlikely to significantly impact our results as ML20 and KM21\_GCM did not show significant differences."*

3. **Comment:** For implementation of both approaches in the box model, there are also many confusing aspects (especially for KM21). Again, some specific issues:
    a. Near lines 181-182, it's stated that phi is set to the maximum AF in the cycle, and does not decrease in subsequent cycles. But this raises the question of how INP are recovered (i.e., how phi is relaxed to the background value phi = 0). This is clarified in the next paragraph on line 188 (phi is to 0 outside cloud), but I would state it in the paragraph around lines 182-183 as otherwise this is confusing.
        i. **Response:** Good point. We updated the text to indicate that it is the phi tracer that is set as the maximum value. The local phi variable changes according to the input to the cirrus sub-model, but is set to zero outside of clouds. This was added to the paragraph.
        ii. **Changes in the text:**

*"As a result, we implemented a tracer in our box model that accounts for $S_i$ oscillations to mimic tracing across GCM timesteps. Following KM21, the tracer ($\phi_{max}$) is set equal to the maximum AF value reached within a cirrus formation cycle. If in the next cycle the $S_i$ is lower than the previous cycle, then no new ice formation can occur until the $S_i$ increases and by extension $\phi_j$ exceeds $\phi_{max}$. Note that outside of a cloud, $\phi$ is set equal to zero."*

b. For implementation in the GCM, presumably phi as well as INP are tracked as prognostic variables and advected and diffused? Of course, there is no transport in the box model, but it should be mentioned how the tracked phi and INP are treated in the full 3D GCM.

    i. **Response:** The phi and INP tracers are only applicable to the cirrus sub-model, which calculates new ice formation based on input from the GCM. We do not apply physical processes like diffusion/advection to these specific tracers. We assume that the changes in phi and INP from the previous timestep by transport are small and do not expect any qualitative impact on our results by neglecting transport of these prognostic variables. We added a note in the main text of the revised manuscript.

    ii. **Changes in the text:**

*"Note as well that in the ECHAM-HAM GCM both $N_{0,i-1}$ and $\phi_{max}$ are not transported."*

c. Relatedly, also on line 181, in the box model it seems that phi is not actually added as a tracer (prognostic variable), but rather ICNC_i-1,j=n (see line 203). Please be very clear about which exact variables are actually tracked in the implementation of these schemes.

    i. **Response:** Phi is traced across GCM timesteps. ICNC is also traced in the GCM but no variable for $ICNC_{i-1,j=n}$ was added to the cirrus box model. This text was reformulated as we excluded the Figure 1 schematic in the revised manuscript in favor of a video supplement that describes the differences between ML20, KM21, and KM21_GCM in more detail. We summarize the video in the revised text.

d. p. 6, eq. 3. This equation is confusing. Why are I and j different indices? Are both time indices? How are they different? Also, line 192 is confusing. I think by "the previous AF" you mean the maximum AF from the previous cycle? If so, please clarify this and use more precise wording. It's also not clear why phi_i-1,j=n has 2 subscripts, when all previous instances of phi have a single subscript. Why? Again, what do I and j actually indicate, and how are they different?

    i. **Response:** "i" is the GCM timestep index and "j" is the cirrus sub-model timestep index. We added some text to make this clearer.

    ii. **Changes in the text:**

*"The cirrus model in ECHAM-HAM is called from the cloud microphysics scheme and calculates the number of new ice crystals that form in in-situ cirrus. It uses a sub-timestepping approach, i.e. within a single GCM timestep (i) there are several sub-timesteps (j) of the cirrus scheme to simulate the temporal evolution of the ice saturation ratio ($S_i$) in an adiabatically ascending air parcel during the formation stage of a cloud (Kuebbeler et al., 2014; Tully et al., 2022c)."*

4. **Comment:** To improve the presentation and flow, I suggest describing the default ML20 approach before the newer KM21 approach. Thus, switch sections 2.1.1 and 2.1.2?
    a. **Response:** We agree. This order of these two sections were rearranged in the revised text.

5. **Comment:** A key limitation of the box model is lack of sedimentation. Around lines 320-322 you mention that some of the large-scale tendencies for Si and INP concentrations in the box model tests may not be realistic, but it seems the lack of ice transport and sedimentation may be a bigger issue when comparing the box model and GCM results. Also see lines 347-350 where it's implied that the biggest differences between the box model and GCM are the magnitude of changes in Si and INP concentrations, not mentioning the role of precipitation generation and sedimentation in the GCM at all. The sentences on lines 376-377 are also relevant here.
    a. **Response:** This was an oversight on our behalf. We added a short discussion to the results section right before Section 3.2 (GCM Simulation) and in the Conclusions that addresses this limitation in our box model.
    b. **Changes in the text:**

*"Some of the changes in large-scale $S_i$ and INP concentrations we tested in the box model were rather extreme in order to examine differences between KM21_GCM and ML20. However, our box model setup is limited as we assume a constant temperature and updraft velocity. We also did not consider other processes such as ice sedimentation, transport, and mixing that would be simulated in a GCM. Furthermore, the KM21 parameterization was developed for a single air parcel within a process model that depicts ice formation within a single cirrus. It does not capture the complexities associated with changes in INP concentrations as well as $S_i$ (among several other factors) across several hundreds of timesteps in a typical GCM simulation. …"*

*"It is important to note that our GCM simulations were also limited as we did not consider transport, vertical diffusion, or ice crystal sedimentation effects on our tracers for the previous INP concentration ($N_{0,i-1}$) and the maximum AF of the previous cirrus cycle ($\phi_{max}$). However, these processes likely would have a small impact on $N_{0,i-1}$ and $\phi_{max}$ and therefore would likely not lead to larger differences between KM21_GCM and ML20."*

6. **Comment:** Line 162. Where does the downdraft part fit it? Does the model include an updraft- downdraft cycle? Or is it only updraft. More description is needed here – this was very confusing.
    a. **Response:** Saturation increases in our cirrus sub-model are fueled by the input vertical velocity. In order to quantify the effect of water vapor deposition onto ice crystals, we calculate a fictious downdraft, which

acts to reduce the updraft and to slow down the increase in saturation if enough deposition has taken place. It works such that if numerous ice crystals form and consume all the available water vapor, then no subsequent ice formation should occur. We included a reference to our previous study (Tully et al., 2022) that uses this same scheme and also includes a detailed description of this method. We reworded this sentence for more clarity.

b. **Changes in the text:**

*"It simulates the temporal evolution of $S_i$ during the adiabatic ascent of a theoretical air parcel. As the $S_i$ is directly related to the updraft velocity (Tully et al., 2022), to quantify the effect of vapor deposition onto newly formed or pre-existing ice crystals a fictitious downdraft is added to the updraft velocity. The resulting net updraft velocity is termed the "effective updraft velocity" and is used to calculate $S_i$."*

7. **Comment:** Lines 220-225. It seems that each cirrus cycle is assumed to have a timescale equal to the GCM time step. This seems like a major assumption and difficult to justify on physical grounds. What is the sensitivity to this? That is, what if there was more than 1 cycle per GCM time step, or less than 1?

   a. **Response:** The cirrus scheme in the GCM is a sub-model that includes its own sub-timestepping. To make this clearer to readers we added text in line with your **Comment 3d** (above).

8. **Comment:** You use term "error" when describing differences between the two approaches. Since there is no benchmark or truth, I feel "difference" would be a better term to use than "error".

   a. **Response:** We use relative error with respect to ML20. A description of our method was added to the Section 2.3 in the revised manuscript following a similar comment by RC1.

**Minor comments**

9. **Comment:** Abstract. Could you explain the differential activated fraction (AF) approach in a simple way while staying within the abstract length limit (e.g., simplify the description on lines 72-75 for here)? This would be useful for readers who aren't familiar with this approach so they can more fully understand the abstract.

   a. **Response:** Yes, a short description was added to the text in the abstract to make this clearer to readers. The description in Lines 72-75 we are happy with, but we did find a way to shorten the sentence. See the changes in the text under **Comment 10**.

10. **Comment:** Line 72. Would it be clearer to say that the differential AF approach describes the number of INPs that are active *per temperature interval*, rather than "within a certain temperature interval" as currently written. Taken literally, the latter does not necessarily imply a differential quantity (i.e., units of IN number per temperature).

    a. **Response:** That is true. We agree with your wording and we changed it in the text.

b. **Changes in the text:**

*"The differential AF approach describes the number of INPs that are active per temperature interval (assuming temperature decreases during a freezing experiment or as a theoretical air parcel rises within a model), whereas the cumulative AF describes the total number of active INPs between the temperature at the onset of ice activity and a given (lower) temperature. This latter quantity equates to the integral of the differential AF over the specified temperature range."*

11. **Comment:** Lines 80-82. I think the term "INP-budgeting" should be "explicit INP-budgeting", in contrast to the "implicit INP-budgeting" in the ML20 approach as mentioned on line in the abstract. Also, not clear why explicit INP budgeting might over-predict ICNC. Can this be explained better?
    a. **Response:** We agree and this was added to the text. This statement was also revised to make it clearer to readers why INP budgeting with the cumulative AF approach may overpredict the number of INPs that activated ice.
    b. **Changes in the text:**

*"For example, if a model budgets INPs by removing them from the total population after they nucleate ice (i.e. explicit INP-budgeting), then using the cumulative AF approach may overpredict the number of heterogeneously nucleated ice crystals (see the example in Section 2.1.2) as it is based on the total number of INPs that could activate between the freezing onset temperature and a given temperature (Vali, 1971, 2019)."*

12. **Comment:** Line 90-91. Can this be explained more? What is the differential ICNC approach, and what is the issue stated by Karcher and Marcolli (2021)?
    a. **Response:** We explain the differential ICNC in more detail in Section 2.1.2. Therefore, we added a reference to this section in the text. We also explicitly referenced the issue stated by Kärcher and Marcolli (2021) for greater clarity.
    b. **Changes in the text:**

*"Note, this new approach in ECHAM-HAM is not the same as the cumulative AF approach described by Kärcher and Marcolli (2021). Muench and Lohmann (2020) introduced implicit INP-budgeting by using a differential ice crystal number concentration (ICNC) variable (Section 2.1.1), which accounts for the issue stated by Kärcher and Marcolli (2021) that using the cumulative AF approach may overpredict the number of ice-active INPs."*

13. **Comment:** Lines 98-99. But according to line 14 in the abstract, GCM simulations were also run and analyzed as part of this study. I'd mention that here.
    a. **Response:** Good point. This was our oversight. We added a reference to the GCM simulations in the text.
    b. **Changes in the text:**

*"In Section 2 we describe the box model we developed based on ECHAM-HAM to analyze these differences. In Section 3 we present the box model results and extend our analysis by presenting results for two simulations in the ECHAM-HAM GCM, followed by a discussion. Finally, we include concluding remarks in Section 4."*

14. **Comment:** Lines 115-119. These two sentences seem to contradict one another. Perhaps in the first sentence, reword to "such that all of the available aerosols that can potentially serve as INPs nucleate ice during a single substep of the cirrus sub-model".

    a. **Response:** Agreed and your wording was adopted in the revised text. Immersion freezing is simulated this way in our model for simplicity. In reality, the coating of mineral dust particles worsens their ability to act as INPs. Therefore, out of all the coated particles, we assume only 5% can act as INPs under cirrus conditions.

15. **Comment:** Line 135. I would replace "should be" with "will be", as "should be" could be mistaken to imply what the concentration would be if there were no error.

    a. **Response:** This is a good point, and we agree. This was changed in the revised text.

16. **Comment:** Line 140. It's not the values per se that are relevant here, but the magnitude of error. Thus, I'd suggest replacing "values" with "error" and then "are not large" with "is not large" for subject-verb agreement.

    a. **Response:** Agreed. It is not the ICNC values, it's the error. We changed this in the revised text.

    b. **Changes in the text:**

*"Although the resulting error between the ICNC values after the second timestep in this short example is not large, not accounting for previously activated INPs in a correct manner could drastically increase the amount of heterogeneous nucleation on mineral dust particles, leading to vastly different cirrus properties."*

17. **Comment:** Line 186. The "tracers" are tracked in time (and time and space in the GCM), so I think it's confusing to say you added a tracer for the "previous INP concentration ($N\_0,i-1$)". Better to say you added a tracer for the INP concentration ($N0$).

    a. **Response:** We need the INP concentration from the previous GCM timestep for our version of the differential AF equation as we need to account for the number of "leftover" INPs. We take away the amount of ice that formed in the previous timestep from this quantity to use in our new KM21_GCM approach. This is explained in greater detail in a video supplement that replaces Figure 1 in the original manuscript. Regarding this sentence, however, we reformulated it for greater clarity.

    b. **Changes in the text:**

*"$S_i$ oscillations are not the only factor to consider across GCM timesteps. INP concentrations can also change. Therefore, we also trace the initial INP concentration ($N_{0,i}$) in our box model. In subsequent timesteps we refer to this quantity as the previous INP concentration ($N_{0,i-1}$)."*

18. **Comment:** Lines 230-234. The description in these sentences is very confusing. The way this is written makes it seem like most cases have small error (< 1%), but this is true for only 9 of the 16 cases. Please rewrite this to make it clearer.

a. **Response:** We reformulated and shortened this paragraph for greater clarity in the revised text.
b. **Changes in the text:**

*"Of the 16 tests we conducted, six show agreement between ML20 and KM21_GCM in the predicted ICNC. For these cases we define agreement as a 0% relative error between KM21_GCM and ML20 for each scenario as denoted by the white shading in Fig. 1. An additional three cases show only very small errors (<1%), indicating that perhaps under most conditions the ML20 and KM21_GCM approaches do not lead to substantially different outcomes. Three additional cases show errors between 1% and 10%. The remaining four cases produce much larger errors (> 17%), which we discuss in more detail below. Note, in all cases with non-zero errors, KM21\_GCM predicts higher ICNC than ML20."*

19. **Comment:** Line 237. It's confusing to say that ML20 only considers the starting INP concentration, because the INP concentration changes over time in most runs (e.g., Table 1). Or do you mean it only considers the initial IN concentration at the start of each cycle? If so, please clarify this.
    a. **Response:** Yes, we meant the initial INP concentration at the start of each cycle. This is in line with your **Comment 17**. We changed "starting" to "initial" in the text.
20. **Comment:** Figure 3. Perhaps label which cases a) and b) are at the top of the plots, rather than only mentioning this in the figure caption.
    a. **Response:** We like this suggestion, and we implemented it for this figure and all related figures in the revised manuscript in line with your **Comments 23** and **25**.
21. **Comment:** Line 298. Not clear what you mean by "with exceptions for non-zero errors". Can you reword this?
    a. **Response:**
    b. **Changes in the text:**

*"While three cases with matching large-scale $S_i$ and INP concentration trends (increasing, decreasing, and constant) showed agreement (Fig. 1), the case where both quantities included an intermediate drop showed a small error of 0.3%. Similarly, all cases with a constant INP concentration showed agreement expect for the case with an intermediate drop in large-scale $S_i$ (0.9%, Fig. 1). As the error for these cases is relatively small, and for brevity within this note, we present the predicted ICNC and $S_i$ profiles in Appendix A."*

22. **Comment:** Line 300. I don't think you mean "error between these cases", but rather the "error for these cases".
    a. **Response:** Correct. This was amended in the revised text.
23. **Comment:** Same comment for Fig. 4 as Fig. 3 above.
    a. **Response:** See response under your **Comment 20**.
24. **Comment:** Lines 336, 341, and elsewhere. What do you mean by the HOM and HET anomalies? Does this just mean the difference in HOM and HET between KM21_GCM and ML20? In which case, I'd replace "anomaly" with "difference" and "anomalies" with "differences" throughout.
    a. **Response:** Yes, that is what we meant by that wording. Relevant wording was amended in the revised text.

25. **Comment:** Appendix. Similar comment for all the figures here as my previous comments for Fig. 3-4.
    a. **Response:** See response under your **Comment 20**.

**Editorial/technical comments**

26. Line 24. I'm not sure what the convention for GMD is, but should acronyms be defined again in the main text even if they're defined first in the abstract? I would lean toward yes.
    a. **Response:** Yes, they should be redefined in the main text. This was addressed in the revised manuscript.
27. Line 29. Remove the second comma.
    a. **Response:** We assume you mean the comma between the two brackets. This was addressed in the revised manuscript such that the acronym and the reference are included in the same set of brackets.
28. Line 44. "that is sparsely populated" seems somewhat awkward wording. Perhaps something like "that has low concentrations"? Similarly, perhaps on line 51 replace "are also sparsely populated" with "are sparse".
    a. **Response:** We agree with your wording and adopted it in the revised manuscript.
29. Line 69. Again, I would define AF here (first time it's used in the main text).
    a. **Response:** This is line with your comment 24 above and was addressed in the revised manuscript.
30. Line 77. Suggest replacing "that can occur" with "occurring".
    a. **Response:** After reviewing this statement with a member of our group who works on ice nucleation in the laboratory, we realized that this statement was incorrect. Plus, it did not add to the discussion. Therefore, we excluded it from the revised manuscript.
31. Line 84. Add "approach" after "AF"?
    a. **Response:** We agree and this sentence was revised in the manuscript
    b. **Changes in the text:**

"*Kärcher and Marcolli (2021) introduced a new parameterization to simulate the number of ice particles resulting from heterogeneous nucleation based on the differential AF approach (Vali, 1971, 2019) while employing INP-budgeting (Section 2.1)*"

32. Line 85. "This method demonstrated that it is able to counteract" is confusing. I think you mean "Karcher and Marcolli (2021) demonstrated that this method is able to counteract..."
    a. **Response:** We partially agree, so we changed *"This method demonstrated…"* to *"They demonstrated…"* in the revised manuscript.
33. Line 125. Add comma before "when".
    a. **Response:** Agreed. This was added to the revised manuscript.
34. Line 128. Change the second "of" to "to" or "in".
    a. **Response:** Thank you for pointing out the wrong use of a preposition. This was addressed.
35. Line 269. I think "produce" should be "produces"?
    a. **Response:** Agreed. This was amended in the revised text.

36. Line 308. Remove comma right before "approach".
    a. **Response:** Agreed. This was removed.
37. Figure A2 caption. Period is missing at the end of the last sentence.
    a. **Response:** Thank you for pointing this out. We added this in.

**Technical Note: assessing predicted cirrus ice properties between two deterministic ice formation parameterizations (EGUSPHERE-2022-1057)**

*Colin Tully, David Neubauer, and Ulrike Lohmann*

**Author Response**

Dear Po-Lun,

Thank you for agreeing to be the editor of our submission to GMD.

We revised the code and data availability statement to the following:

*"The ECHAM-HAMMOZ model is freely available to the scientific community under the HAMMOZ Software License Agreement, which defines the conditions under which the model can be used (https://redmine.hammoz.ethz.ch/projects/hammoz/wiki/2_How_to_get_the_sources, last access: 19 October 2022). The specific version of the code used for this study is archived in the ECHAM-HAMMOZ SVN repository at https://svn.iac.ethz.ch/external/echam-hammoz/echam6-hammoz/tags/papers/2022/Tully_et_al_2022_GMD_for-review (last access: 21 October 2022). More information can be found on the HAMMOZ website (https://redmine.hammoz.ethz.ch/projects/hammoz, (last access: 19 October 2022). The box model that is based on the ECHAM-HAM code that was used to produce the heterogeneous nucleation-only plots in this manuscript, as well as other post-processing and analysis scripts are archived on Zenodo (Tully et al., 2022b). The processed GCM output data to produce the relevant plots in this manuscript are also available on Zenodo (Tully et al., 2022a)."*

Both Zenodo links are open access. We hope this addresses your comments.

Sincerely,
Colin Tully (on behalf of all co-authors)

**References**

- Tully, C., Neubauer, D., and Lohmann, U.: Data for the "Technical Note: assessing predicted cirrus ice properties between two deterministic ice formation parameterizations" manuscript, https://doi.org/10.5281/zenodo.7125683, 2022a.
- Tully, C., Neubauer, D., and Lohmann, U.: Data analysis and plotting scripts for the "Technical Note: assessing predicted cirrus ice properties between two deterministic ice formation parameterizations" manuscript, https://doi.org/10.5281/zenodo.7234344, 2022b.

Dear Juan,

Thank you for providing more information on your specific concerns.

The ECHAM-HAMMOZ license, which includes the ECHAM license, forbids public distribution of the code. This is out of our control as the ECHAM model is part of the Max Plank Institute (MPI) Earth System Model (ESM), which is available through a license (https://mpimet.mpg.de/en/science/models/availability-licenses).

Here I quote the Preamble of the ECHAM-HAMMOZ license agreement for greater clarity:

*"ECHAM-HAMMOZ is a state-of-the-art chemistry climate model based on the ECHAM general circulation model which has been developed by the Max Planck Institute for Meteorology (MPI-M) in Hamburg, Germany. ECHAM-HAMMOZ contains many complex interactions between the physical and biogeochemical processes in the Earth system and is intended for the analysis of atmospheric processes and decadal-scale assessment studies investigating the potential impacts of changing emissions or the relevance of atmospheric composition changes for climate change and ecosystems. The chemistry and aerosol modules and the feedback processes between chemistry and climate are additions to the original ECHAM model and have been jointly developed in a consortium composed of Eidgenössisch Technische Hochschule Zürich (ETHZ), Max Planck Institut für Meteorologie, Hamburg, Forschungszentrum Jülich GmbH, University of Oxford, and the Finnish Meteorological Institute (copyright holders). ECHAM-HAMMOZ thus is a separate software package from the original ECHAM model, but a version of ECHAM itself is an integral part of ECHAM-HAMMOZ. The mentioned copyright holders of the ECHAM-HAMMOZ software package agreed that this ECHAM-HAMMOZ license includes the license for ECHAM (see ECHAM software package or http://www.mpimet.mpg.de/en/science/models/model- distribution/licence.html) and have authorized ETHZ to license the entire software package within the scope of this agreement."*

As mentioned in our previous response the version of the code used in this study is available on Zenodo here: https://zenodo.org/record/7610091 (DOI: 10.5281/zenodo.7610090). The retention period of the version of the code used in this study is therefore at least 20 years (https://about.zenodo.org/policies/). As mentioned in our previous response we created a private link that is available for the review process that bypasses the restricted access. Please contact me directly so I can send you this link so you can use it for editorial purposes and to send it to reviewers who need access to the model code. Please note that material is subject to strict confidentiality. The private link will expire at the end of the review process, and no one should distribute or keep this material obtained through this private link.

Best regards,
Colin Tully (on behalf of all co-authors)

---

## Referee Report (RR1)

Second review of "**Technical note: Assessing predicted cirrus ice properties between two deterministic ice formation parameterizations**", by Tully et al., submitted *GMD*.

Note: line numbers referred to below correspond to the track changed version.

**General comments.** The paper is substantially improved from the previous version, particularly with regard to the description of the KM21 and ML20 approaches and their implementation in the box model and GCM. I still have a handful of minor comments and suggestions below, and also point out a few places where the descriptions are still confusing (e.g., the "fictitious downdraft"). My recommendation is to accept the paper pending these minor revisions.

I will also note the authors said in their replies to previous comments that they will provide a video supplement replacing Fig. 1, which further describes the approaches. This was not available at the time of my review, so I can't comment on it. That said, I think the description of the approaches in the current revised manuscript is sufficient for understanding by readers.

**Overall recommendation**: *Minor revision*

**Minor and editorial comments.**

Lines 12-13, abstract. "over-prediction of the importance of heterogeneous nucleation within cirrus" seems awkwardly worded. Would "over-prediction of heterogeneous nucleation within cirrus be better"?

Line 26, abstract. Remove "and".

Line 74. Grammar → the sentence starting here doesn't seem like a complete sentence.

Line 92. Should "saturation" be "supersaturation"?

Line 121. I'd replace "for" with "to understand".

Line 147. I don't understand where you say "Threshold mechanisms are based on the stochastic nature of nucleation rates", whereby all particles that can potentially nucleate under given conditions do so when the threshold is crossed. But wouldn't a stochastic process imply there is not a such a threshold, and instead there is a fraction of INPs that actually nucleate ice over some time interval? I would think the stochastic nature of nucleation rates would imply a continuous process.

Lines 165-166. You say that the activation of INPs during the lifetime of a cirrus is implicitly included by requiring $ICNC_j > ICNC_{j-1}$. Does this mean there are no sink processes for ICNC considered during the cirrus sub-steps (from, e.g. sedimentation). Also, I think there should be a greater than or equal to sign here, rather than greater than, since it's possible that no further nucleation occurs during a cirrus sub-timestep.

Line 169. I don't think you need to redefine $\phi$ here since it's already used in Eq. 1 and defined as the cumulative AF.

Lines 176-185. The example shown here to illustrate over-prediction of ICNC for the cumulative AF approach is much clearer than the previous version of the manuscript. However, it would still be good to clarify why $\phi$ = 0.05 in the first step and 0.1 in the second step. Is this simply taking the total AF (0.1) and dividing half of it into the first cirrus substep?

Line 185. Why not be more precise in this example for $\psi_j$, where it's equal to ~0.0526? When this is multiplied by 95 L$^{-1}$, it gives very close to 5 L$^{-1}$. I don't think you need to give $\psi_j$ approximately as 0.05.

Lines 208-212. The "fictitious downdraft" in the box cirrus model still does not make sense to me. In the authors' reply to my previous comment about this, they state that such a downdraft "acts to reduce the updraft and slow down the increase in saturation if enough deposition has taken place". It's not clear what's meant by "enough deposition". Enough relative to what? It seems the vertical velocity (updraft and downdraft) should only be an input to the cirrus model, and it's not clear why the vertical velocity should be modified somehow (by adding downdraft, thus effectively reducing updraft) to limit deposition.

Line 243. Perhaps to be clearer, you could say the tracer is not "advected or diffused" in the GCM implementation, rather than simply "not transported".

Figure 1 and lines 296-305. It's still not exactly clear how this error is calculated. In the figure caption, "maximum relative error between KM21_GCM and ML20" isn't clear. Is this error relative to ML20, or to KM21_GCM. In other words, is this calculated as (KM21_GCM – ML20)/ML20? Or (ML20 – KM21_GCM)/KM21_GCM? Or something else? Giving the exact equation used to calculate the error would clear up any confusion.

Line 422. I don't think you want to call ML20 the "reference" case, since that implies this is a ground truth. Thus, I feel it's better to call this the "control" case. Also, throughout this section you refer to "differences", are these relative to ML20 or KM21_GCM? I think the former, but it would be good to make this very clear.

Line 431. Perhaps "just north of the equator" rather than "just above the equator"?

Figure 4. In this section you've changed all usage of "anomaly" to "difference" in the text, which is an improvement (I had suggested it in the previous review). However, the plot titles in b) and d) still say "Anomaly", so I suggest changing this to "Difference" for consistency with the text and figure caption.

Line 441. I don't think "SH" is defined yet as Southern Hemisphere.

While it's clear that the fmax and N0 tracers for the KM21_GCM approach are not advected or diffused when implemented in the GCM, are the HET and HOM ICNC tracers advected/diffused in the GCM tests? I would assume so, since ICNC itself is advected/diffused in the model (I think). This should be clarified in section 3.2.

Lines 518-519. I think this sentence is a bit confusing, suggest rewording it to: "In our model, it is assumed that each ice particle, including snow crystals, includes a single INP, where in reality there may be numerous INPs associated with snow crystal aggregates composed of several ice crystals."

Line 519. For ice crystals, the more common term for this process is aggregation rather than collision-coalescence. Thus, I suggest replacing "collision and coalescence process" with "collision and aggregation process".

---

## Author Response (AR2)

**Technical Note: assessing predicted cirrus ice properties between two deterministic ice formation parameterizations (egusphere-2022-1057)**

*Colin Tully, David Neubauer, and Ulrike Lohmann*

**Topical Editor Decision Author Response**

Dear Po-Lun,

Thank you very much for taking time to serve as the topical editor for our submission to GMD.

I have quoted each of the reviewer's comments below with our responses and changes in the text where applicable.

Please be aware that some of the lines that are quoted in the comments do not align with the lines in the manuscript. We tried to match the comments with the relevant lines as much as possible.

Sincerely,
Colin Tully (on behalf of all co-authors)

**Comments**

1. **Comment:** Lines 12-13, abstract. "over-prediction of the importance of heterogeneous nucleation within cirrus" seems awkwardly worded. Would "over-prediction of heterogeneous nucleation within cirrus be better"?
   a. **Response:** We agree, and we changed the phrase in the text with your suggested wording.
   b. **Changes in the text:**

*"They argued that this new approach with explicit INP-budgeting, which removes INPs from the total population after they nucleate ice, could help to correct a potential over-prediction of heterogeneous nucleation within cirrus when budgeting is not considered."*

2. **Comment:** Line 26, abstract. Remove "and".
   a. **Response:** Thank you for finding that typo. This was removed.

3. **Comment:** Line 74. Grammar: the sentence starting here doesn't seem like a complete sentence.
   a. **Response:** Is this the line starting with "Generally, the factors discussed above"? We reworded this sentence in the revised manuscript.
   b. **Changes in the text:**

*"Generally, the factors discussed above lead to an overall poor predictability of how INPs influence heterogeneous nucleation mechanisms in cirrus and they contribute to uncertainties when simulating these mechanisms in numerical models."*

4. **Comment:** Line 92. Should "saturation" be "supersaturation"?
   a. **Response:** If this is referring to the "ice saturation ratio" then no, it should read as such as that is the established name of that variable. It's value > 1.0 implies supersaturation.

5. **Comment:** Line 121. I'd replace "for" with "to understand".
   a. **Response:** We agree, and we revised the text with your wording.
   b. **Changes in the text:**

*"In this note, we utilize the box model to compare a GCM-compatible differential AF approach based on Kärcher and Marcolli (2021) to understand heterogeneous nucleation to the AF approach by Muench and Lohmann (2020), hereafter abbreviated as ML20."*

6. **Comment:** Line 147. I don't understand where you say "Threshold mechanisms are based on the stochastic nature of nucleation rates", whereby all particles that can potentially nucleate under given conditions do so when the threshold is crossed. But wouldn't a stochastic process imply there is not a such a threshold, and instead there is a fraction of INPs that actually nucleate ice over some time interval? I would think the stochastic nature of nucleation rates would imply a continuous process.
   a. **Response:** This wording was incorrect on my behalf. We reworded this paragraph to make it clearer and to fix my mistake.
   b. **Changes in the text:**

*"Muench and Lohmann (2020) distinguish between two approaches (a threshold approach and a continuous approach) for heterogeneous nucleation within cirrus. For the threshold approach, as soon as the $S_i$ reaches a critical value (i.e. a threshold), the model assumes nucleation rates are efficient enough such that all of the available aerosols that can potentially serve as INPs nucleate ice during a single step of the cirrus sub-model. For immersion freezing of internally mixed mineral dust particles, it is assumed that only 5% of the background concentration can act as INPs (Gasparini and Lohmann, 2016; Muench and Lohmann, 2020). Heterogeneous deposition nucleation, on the other hand, is based on laboratory measurements of AF by Möhler et al. (2006), which are determined by temperature (T) and $S_i$ and which increases continuously with decreasing T and increasing $S_i$. In the cirrus sub-model, this approach is applied to deposition nucleation onto externally mixed (insoluble) mineral dust particles only."*

7. **Comment:** Lines 165-166. You say that the activation of INPs during the lifetime of a cirrus is implicitly included by requiring ICNCj > ICNCj-1. Does this mean there are no sink processes for ICNC considered during the cirrus sub-steps (from, e.g. sedimentation). Also, I think there should be a greater than or equal to sign here, rather than greater than, since it's possible that no further nucleation occurs during a cirrus sub-timestep.

a. **Response:** Correct. there are no sink terms for ice within the cirrus sub-model. It only calculates new ice formation. You are right about the greater than or equal to sign. We changed this in the text.

8. **Comment:** Line 169. I don't think you need to redefine "phi" here since it's already used in Eq. 1 and defined as the cumulative AF.
   a. **Response:** Is this referring to this sentence: "In the first instance of ice formation in the current cirrus cycle (current GCM timestep), the leftover INPs nucleate ice according to the differential AF ($\psi_j$) and the newly available INPs nucleate ice according to the cumulative AF ($\phi_j$) as denoted in the numerator of Equation 3"? If so, we agree and we used the symbols to describe each AF approach in the revised text.

9. **Comment:** Lines 176-185. The example shown here to illustrate over-prediction of ICNC for the cumulative AF approach is much clearer than the previous version of the manuscript. However, it would still be good to clarify why phi = 0.05 in the first step and 0.1 in the second step. Is this simply taking the total AF (0.1) and dividing half of it into the first cirrus substep?
   a. **Response:** No, it is simply an explanation of how it works. The exact values in this case do not make a difference as we are explaining KM21's argument. We revised the text to make this clearer that these values are simply assumptions.
   b. **Changes in the text:**

*"As a short conceptual example of their argument (see also KM21 Figure 1), let's assume two cirrus model timesteps starting from $\phi_{j=0}$ = 0. In the first cirrus model timestep $\phi_{j=1}$ = 0.05 and in the second cirrus model timestep $\phi_{j=2}$ = 0.1 under ambient temperature and $S_i$."*

10. **Comment:** Line 185. Why not be more precise in this example for "psi", where it's equal to ~0.0526? When this is multiplied by 95 L^-1, it gives very close to 5 L^-1. I don't think you need to give psi approximately as 0.05.
    a. **Response:** We somewhat agree that this should be clearer, so we added one more digit to the text. It now reads as "0.053".

11. **Comment:** Lines 208-212. The "fictitious downdraft" in the box cirrus model still does not make sense to me. In the authors' reply to my previous comment about this, they state that such a downdraft "acts to reduce the updraft and slow down the increase in saturation if enough deposition has taken place". It's not clear what's meant by "enough deposition". Enough relative to what? It seems the vertical velocity (updraft and downdraft) should only be an input to the cirrus model, and it's not clear why the vertical velocity should be modified somehow (by adding downdraft, thus effectively reducing updraft) to limit deposition.
    a. **Response:** The vertical velocity is an input variable to the cirrus sub-model, which is used to determine the cooling rate of the adiabatic ascent of an air parcel, which in turn determines the degree of ice supersaturation. Therefore, we must have a way to counteract Si

increasing if water vapor has been consumed by deposition onto an INP or ice crystals. We quantify this consumption as a "fictitious downdraft" that is only ever used to update the vertical velocity at every time step in the cirrus sub-model in order to simulate the effect of a "deceleration" of Si increasing. If a sufficient amount of new ice has formed or if there is a large concentration of pre-existing ice crystals, then deposition will consume all available water vapor and the fictitious downdraft will outweigh the updraft and prevent further ice formation from occurring within the cirrus sub-model. The modified vertical velocity is only used to compute new ice formation in the cirrus model. For the deposition of water vapor onto INPs or ice crystals the unmodified vertical velocity is used. We added a short statement in the revised text to make this clear.
   b. **Changes in the text:**

*"The resulting net updraft velocity is termed the "effective updraft velocity" and is used to calculate $S_i$ (note that the original updraft velocity is used to compute vapor deposition onto newly formed or pre-existing ice crystals)."*

12. **Comment:** Line 243. Perhaps to be clearer, you could say the tracer is not "advected or diffused" in the GCM implementation, rather than simply "not transported".
   a. **Response:** We agree and your wording was used in the revised text.
   b. **Changes in the text:**

*"Note as well that in the ECHAM-HAM GCM both $N_{0,i-1}$ and $\phi_{max}$ are not advected or diffused."*

13. **Comment:** Figure 1 and lines 296-305. It's still not exactly clear how this error is calculated. In the figure caption, "maximum relative error between KM21_GCM and ML20" isn't clear. Is this error relative to ML20, or to KM21_GCM. In other words, is this calculated as (KM21_GCM – ML20)/ML20? Or (ML20 – KM21_GCM)/KM21_GCM? Or something else? Giving the exact equation used to calculate the error would clear up any confusion.
   a. **Response:** Good point. An equation was added to the text to explicitly define this.
   b. **Changes in the text:**

*"We assess each case by the relative error between the KM21_GCM approach and the ML20 approach, according to Equation 4. Finally, we conducted two simulations with the ECHAM-HAM GCM to compare ICNC fields and cloud properties between ML20 and KM21_GCM."*

$$Error = \frac{ICNC_{KM21\_GCM} - ICNC_{ML20}}{ICNC_{ML20}} x100$$

14. **Comment:** Line 422. I don't think you want to call ML20 the "reference" case, since that implies this is a ground truth. Thus, I feel it's better to call this the "control" case. Also, throughout this section you refer to "differences", are these relative to ML20 or KM21_GCM? I think the former, but it would be good to make this very clear.

    a. **Response:** This is a good point. Thank you. We found two instances of this and changed both in the revised text. Regarding the differences, as this is related to your **Comment 13**, we feel the new equation 4 addresses this issue.

15. **Comment:** Line 431. Perhaps "just north of the equator" rather than "just above the equator"?

    a. **Response:** We agree and this was revised in the text.

    b. **Changes in the text:**

*"In the tropics, just north of the equator, we find that KM21_GCM produces less HET than ML20, which corresponds to an increase in HOM around the same region."*

16. **Comment:** Figure 4. In this section you've changed all usage of "anomaly" to "difference" in the text, which is an improvement (I had suggested it in the previous review). However, the plot titles in b) and d) still say "Anomaly", so I suggest changing this to "Difference" for consistency with the text and figure caption.

    a. **Response:** Thank you for finding that. We changed this in the revised manuscript.

17. **Comment:** Line 441. I don't think "SH" is defined yet as Southern Hemisphere.

    a. **Response:** Thank you for pointing this out. We defined the acronym in the revised text.

18. **Comment:** While it's clear that the fmax and N0 tracers for the KM21_GCM approach are not advected or diffused when implemented in the GCM, are the HET and HOM ICNC tracers advected/diffused in the GCM tests? I would assume so, since ICNC itself is advected/diffused in the model (I think). This should be clarified in section 3.2.

    a. **Response:** We agree, and we added a short statement to clarify.

    b. **Changes in the text:**

*"Like Tully et al. (2022c, these ICNC tracers are advected and diffused every GCM timestep. Similarly, we calculate a 5% significance based on the inter-annual variability of the five-year simulations. For the remainder of this section, we base significance on this method."*

19. **Comment:** Lines 518-519. I think this sentence is a bit confusing, suggest rewording it to: "In our model, it is assumed that each ice particle, including snow crystals, includes a single INP, where in reality there may be numerous

INPs associated with snow crystal aggregates composed of several ice crystals."

    a. **Response:** We agree and we implemented slightly revised wording in the text.

    b. **Changes in the text:**

*"In our model, it is assumed that each ice crystal, including snow crystals, includes a single INP, whereas in reality there may be numerous INPs associated with snow crystal aggregates composed of several ice crystals."*

20. **Comment:** Line 519. For ice crystals, the more common term for this process is aggregation rather than collision-coalescence. Thus, I suggest replacing "collision and coalescence process" with "collision and aggregation process".

    a. **Response:** We agree completely. This was our oversight. Thank you for point that out. We changed this in the revised text.

    b. **Changes in the text:**

*"Therefore, by not considering collision and aggregation processes, our model may underestimate the number of previously formed ice crystals."*